# Astrocytic ALKBH5 in stress response contributes to depressive-like behaviors in mice

Fang Guo[1,7], Jun Fan[2,7], Jin-Ming Liu[1,7], Peng-Li Kong[1], Jing Ren[1], Jia-Wen Mo[1], Cheng-Lin Lu[1], Qiu-Ling Zhong[1], Liang-Yu Chen[1], Hao-Tian Jiang[1], Canyuan Zhang[1], You-Lu Wen[3], Ting-Ting Gu[3], Shu-Ji Li[1], Ying-Ying Fang[1], Bing-Xing Pan [4], Tian-Ming Gao [1] & Xiong Cao [1,5,6] ✉

Epigenetic mechanisms bridge genetic and environmental factors that contribute to the pathogenesis of major depression disorder (MDD). However, the cellular specificity and sensitivity of environmental stress on brain epitranscriptomics and its impact on depression remain unclear. Here, we found that ALKBH5, an RNA demethylase of N6-methyladenosine (m6A), was increased in MDD patients' blood and depression models. ALKBH5 in astrocytes was more sensitive to stress than that in neurons and endothelial cells. Selective deletion of ALKBH5 in astrocytes, but not in neurons and endothelial cells, produced antidepressant-like behaviors. Astrocytic ALKBH5 in the mPFC regulated depression-related behaviors bidirectionally. Meanwhile, ALKBH5 modulated glutamate transporter-1 (GLT-1) m6A modification and increased the expression of GLT-1 in astrocytes. ALKBH5 astrocyte-specific knockout preserved stress-induced disruption of glutamatergic synaptic transmission, neuronal atrophy and defective Ca$^{2+}$ activity. Moreover, enhanced m6A modification with S-adenosylmethionine (SAMe) produced antidepressant-like effects. Our findings indicate that astrocytic epitranscriptomics contribute to depressive-like behaviors and that astrocytic ALKBH5 may be a therapeutic target for depression.

Major depressive disorder (MDD) is one of the most common psychiatric diseases across the world, affecting 15–18% population all through their lives[1,2]. It is critical for public health for its serious consequences due to its high prevalence, high recurrence rates, chronicity, comorbidity with physical illness, and leading disease burden[1–3]. The onset and development of depression result from a combination of genetic and environmental factors, with stress being a high-risk factor[4,5]. Mechanistically, epigenetic change in brain cells

[1]Key Laboratory of Mental Health of the Ministry of Education, Guangdong-Hong Kong-Macao Greater Bay Area Center for Brain Science and Brain-Inspired Intelligence, Guangdong-Hong Kong Joint Laboratory for Psychiatric Disorders, Guangdong Province Key Laboratory of Psychiatric Disorders, Guangdong Basic Research Center of Excellence for Integrated Traditional and Western Medicine for Qingzhi Diseases, Department of Neurobiology, School of Basic Medical Sciences, Southern Medical University, Guangzhou, China. [2]Department of Anesthesia, Guangzhou Women and Children's Medical Center, Guangzhou Medical University, Guangdong Provincial Clinical Research Center for Child Health, Guangzhou, Guangdong, China. [3]Department of Psychology and Behavior, Guangdong 999 Brain Hospital, Institute for Brain Research and Rehabilitation, South China Normal University, Guangzhou, Guangdong, P. R. China. [4]Department of Biological Science, School of Life Science, Nanchang University, Nanchang, China. [5]Department of Oncology, Nanfang Hospital, Southern Medical University Guangzhou, Guangdong, P. R. China. [6]Microbiome Medicine Center, Department of Laboratory Medicine, Zhujiang Hospital, Southern Medical University, Guangzhou, Guangdong, P. R. China. [7]These authors contributed equally: Fang Guo, Jun Fan, Jin-Ming Liu. ✉e-mail: caoxiong@smu.edu.cn

subject to environmental stress impacts functions to exacerbate depressive symptoms[4]. However, the cellular specificity and sensitivity of stress effects on brain epitranscriptomics, and their impact on depression remain largely unknown.

N6-methyladenosine (m6A) methylation, as the most abundant internal modification of eukaryotic mRNA, is reported to regulate transcript processing and translation[6]. Mammalian m6A regulates postnatal growth in the mouse brain, memory processes in the prefrontal cortex and the hippocampus, and adult neurogenesis[7]. Recent studies have found that global m6A methylation in human and mouse whole blood is temporarily decreased following acute stress[8]. RNA methyltransferases (METTL3, METTL14, and WTAP) and demethylases (FTO and ALKBH5) are altered in the MDD patients and the mouse models of depression[9]. And deletion of METTL3 or FTO in adult hippocampal excitatory neurons increases fear memory[8] and mood disorders-related behaviors[9], suggesting m6A/m-RNA methylation is involved in the pathogenesis of depression.

Astrocytes, major cell types of glial cells, are the key components of the blood-brain barrier (BBB)[10] and also wrap around neuronal synapses to form tripartite synapse structures[11,12]. When stress occurs, astrocytes can preferentially sense the stress factors from plasma than neurons[13,14], modulate neuronal transmission, and participate in the regulation of stress response[15]. In addition, astrocytic dysfunction has been linked to the pathophysiology of depression in animal studies and postmortem brain analyses of patients with depression[16–19]. Together, these studies raise potential concerns that astrocytic epitranscriptome is more sensitive to environmental stressors and mediates depressive-like behaviors.

Here, we reported that the ALKBH5, a well-characterized RNA demethylase, was increased in patients with MDD and mouse models of depression. ALKBH5 in astrocytes was sufficient to bidirectionally regulate susceptibility to stress, but not in neurons and endothelial cells. Astrocytic ALKBH5 in the mPFC modulated glutamatergic transmission through the m6A RNA methylation of glutamate transporter-1 (GLT-1). Combining neuronal morphometric analysis and fiber-photometry, we found that astrocytic ALKBH5 maintained neuronal morphology and calcium activity under chronic stress. Furthermore, the supplementation of S-adenosylmethionine (SAMe), used to sustain m6A modification, produced antidepressant-like behaviors. In conclusion, our findings reveal that astrocytic ALKBH5, as a regulator of depressive-like behaviors, is more sensitive to environmental stressors.

## Results

### The expression of ALKBH5 is upregulated both in MDD patients and in mouse models of depression

To investigate whether the m6A-modifying is aberrant in depression, we performed real-time quantitative PCR (qRT-PCR) analysis of the peripheral blood of 30 MDD patients and 36 healthy controls. The results showed that ALKBH5 was significantly upregulated and FTO was downregulated in the patient group among RNA methyltransferases (METTL3, METTL14, and WTAP), demethylases (FTO and ALKBH5) and m6A binding protein (YTHDF1 and YTHDF2) (Fig. 1a). Similarly, we reanalyzed published MDD datasets (GSE102556), and the results revealed that mRNA expression of ALKBH5 was significantly increased in the dlPFC of MDD patients compared with the healthy controls (Fig. 1b)[20]. Next, we used chronic social defeat stress (CSDS) to mimic depressive-like behavior in preclinical studies of depression[21]. In the 10 days CSDS, 8-week-old male C57BL/6J male mice were exposed to an aggressive CD-1 mouse for social defeat and separated into susceptible (Sus) and resilient (Res) subpopulations as assessed with the social interaction (SI) test (Supplementary Fig. 1a, b). After CSDS, we collected the peripheral blood and the tissues of mPFC, a candidate site for impaired function in depression[13,19], and examined the mRNA expression of m6A-modifying enzymes. The mRNA expression of

ALKBH5 in the peripheral blood of Sus mice was significantly upregulated, whereas the RNA expression levels of METTL3 and FTO were downregulated both in Sus and Res mice compared with control (Ctrl) mice (Fig. 1c). Besides, the ALKBH5 and FTO mRNA were also upregulated in the mPFC of Sus mice, whereas the METTL3 and METTL14 levels were changed in both of Sus and Res mice (Fig. 1d). Notably, ALKBH5 mRNA levels were increased in human and mouse peripheral blood as well as in brain tissues; and the ALKBH5 mRNA levels in the mPFC of Ctrl, Sus and Res mice correlated with SI Ratio in the SI test, respectively (Fig. 1e). Thus, we directly analyzed ALKBH5 protein levels in several mood-related brain regions, including the mPFC, nucleus accumbens (NAc), amygdala (Amy), striatum (Stri), hippocampus (Hip) and dorsal raphe nucleus (DRN) after CSDS. Western blotting analysis showed that ALKBH5 protein was specifically increased in the mPFC of Sus mice (Fig. 1f and Supplementary Fig. 1d), and the ALKBH5 protein level was correlated with SI ratio (Supplementary Fig. 1c). Furthermore, the total m6A level was significantly decreased in the mPFC of Sus mice rather than Ctrl mice (Fig. 1g). In addition, to determine whether stressed female mice also showed such increases in ALKBH5 expression, we used a protocol for CSDS in female mice. Following the CSDS paradigm, mPFC tissue was isolated from the mice and used for qPCR analysis. The result showed that ALKBH5 in the mPFC of female Sus mice was also increased, as in male mice (Supplementary Fig. 1e). In the following, we measured the expression of ALKBH5 in lipopolysaccharide (LPS)-induced mouse model of depression[22]. A total of 7 days LPS treatment (0.5 mg kg⁻¹ day⁻¹, intraperitoneal) induced a depressive-like phenotype in the adult male C57BL/6J mice in the forced swim test (FST) (Supplementary Fig. 1f). In the LPS-treated mice, ALKBH5 and FTO levels in the mPFC were also observed to have significantly increased (Fig. 1h and Supplementary Fig. 1g). These results suggest that ALKBH5 levels are consistently elevated in human and mouse peripheral blood and brain tissue.

Neurons, glial cells, and endothelial cells are representative cell types in the adult brain[23]. To identify ALKBH5 in which subtype of cells responds primarily to stress, we detected ALKBH5 protein levels in vitro and in vivo. First, the GR agonist dexamethasone (DXMS) or LPS was applied in a dose- and time-dependent manner to cultured neurons, endothelial cell line (bEnd3), or astrocytes representing glial cells. Western blotting showed an increase in ALKBH5 protein levels in astrocytes treated with low concentrations of DXMS (1 μM) or LPS (0.5 μg mL⁻¹), and neurons showed an increase in ALKBH5 protein levels only when treated with high concentrations of DXMS (5 μM) or LPS (1.5 μg mL⁻¹), whereas endothelial cells showed no significant change in ALKBH5 protein in response to DXMS or LPS treatment (Fig. 1i, j). Meanwhile, the upregulation of ALKBH5 in astrocytes was faster than that in neurons and endothelial cells when treated with DXMS (1 μM) or LPS (1 μg mL⁻¹), suggesting that ALKBH5 in glial cells is more sensitive to stress than that in neurons and endothelial cells (Supplementary Fig. 2a, b). Furthermore, to confirm the change of glial ALKBH5 in vivo, astrocytes, microglia, or oligodendrocytes in the mPFC were collected by fluorescence-activated cell sorting (FACS) or Magnetic cell sorting (MACS) after CSDS (Supplementary Fig. 1h). qRT-PCR analysis showed that ALKBH5 was decreased in microglia in the mPFC of Sus mice and ALKBH5 was no significant change in oligodendrocytes (Supplementary Fig. 3a, b). Meanwhile, simple western blotting analysis showed that astrocytic ALKBH5 was significantly increased in the mPFC of Sus mice, consistent with the change in total ALKBH5 in Fig. 1 (Supplementary Fig. 1i). Together, these results suggest that the elevation of astrocytic ALKBH5 is a major contributor to the upregulation of total ALKBH5 in animal models.

### Astrocyte-specific knockout of ALKBH5 produces antidepressant-like behaviors

To examine how ALKBH5 affects depressive-related phenotypes, we generated astrocyte-specific, neuron-specific, and endothelial cell-

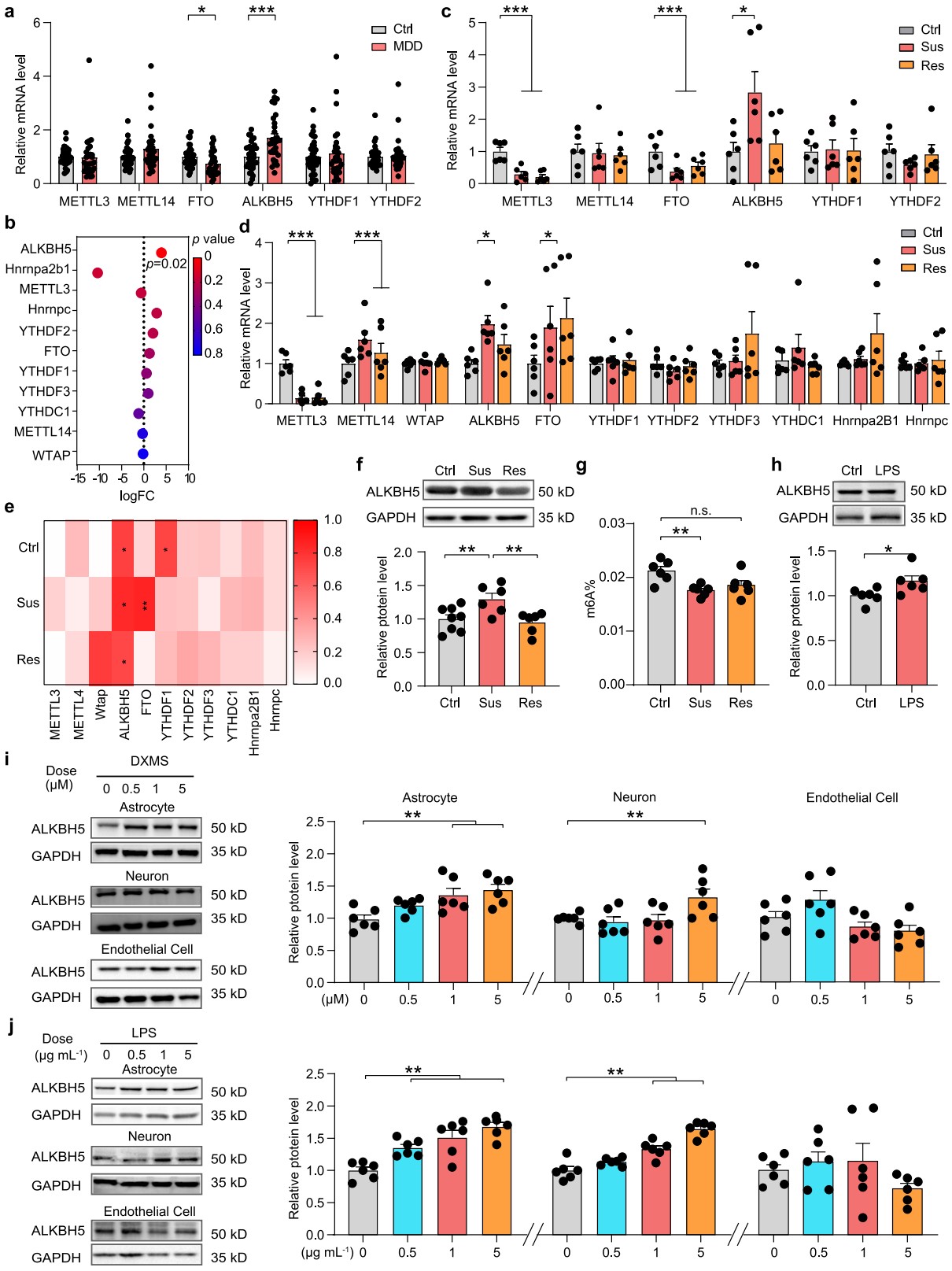

specific conditional ALKBH5 deletion mouse lines by crossing the *ALKBH5* allele in which *loxP* sites flank exon 1 with *Fgfr3-iCreER^{T2}*, *CaMKIIα-creER^{T2}* or *Cdh5-iCreER^{T2}* transgenic lines (Fig. 2a and Supplementary Fig. 4a). Astrocyte cKO, Neuron cKO, EC cKO and littermate Ctrl mice exhibited no changes respectively in growth rate, body weight, brain size as well as in brain structure (Supplementary Figs. 4b–d, 6a–c, 7a). Western blotting analysis showed that ALKBH5

protein was significantly decreased in Astrocyte cKO mice (Supplementary Fig. 4f) and Neuron cKO (Fig. 2h) mice after treatment with tamoxifen. Immunostaining revealed that ALKBH5 were attenuated in S100β positive astrocytes in the mPFC of Astrocyte cKO mice (Fig. 2b, c and Supplementary Fig. 4e), NeuN positive neurons in the mPFC of Neuron cKO mice (Fig. 2i and Supplementary Fig. 6d) and Cdh5 positive endothelial cells in the mPFC of EC cKO mice (Fig. 2m, n)

**Fig. 1 | ALKBH5 expression is upregulated in MDD patients and mouse depression models. a** Real-time quantitative PCR (qRT-PCR) analysis of the genes encoding m6A-modifying enzymes in the peripheral blood of MDD patients and healthy individuals (Ctrl). (Ctrl, $n = 36$; MDD, $n = 30$ participants). **b** Gene expression (logFC) in the dlPFC of MDD compared with linear regression. **c** qRT-PCR analysis of the genes encoding m6A-modifying enzymes in the peripheral blood of Sus, Res, and Ctrl mice. ($n = 6$ mice per group). **d** qRT-PCR analysis of the genes encoding m6A-modifying enzymes in the mPFC of Sus, Res, and Ctrl mice. ($n = 6$ mice per group). **e** Several patterns of gene expression correlated with the SI ratio. ($n = 6$ mice per group). **f** ALKBH5 protein in the mPFC of mice after CSDS. (Ctrl, $n = 8$; Sus, $n = 6$; Res, $n = 6$ mice). **g** Quantification of m6A levels in mRNA of the mPFC of Sus, Res, and Ctrl mice. ($n = 6$ mice per group). **h** ALKBH5 protein in the mPFC of mice with LPS-induced depression. ($n = 6$ mice per group). **i, j** Western blotting analysis of ALKBH5 protein in the primary cultured astrocytes (DIV8), neurons (DIV14) and endothelial cell line (bEnd3) treated with DXMS (0, 0.5, 1, and 5 μM) (**i**) or LPS (0, 0.5, 1, and 5 μg mL$^{-1}$) (**j**) for 4 h. (**i, j**, $n = 6$ wells per group). Data were presented as the mean ± SEM. Two-sided unpaired $t$-test (**a, h**) or one-way ANOVA followed by Bonferroni's test for multiple comparisons test (**c, d, f, g, i, j**). Two-tailed Fisher's exact test and FDR corrected $p < 0.05$ were considered significant (**b**). *$p < 0.05$; **$p < 0.01$; ***$p < 0.001$; n.s. no significance. Source data are provided as a Source Data file. See Supplementary Data 4 for statistical details.

compared with Ctrl mice. We co-stained ALKBH5 and NeuN, Iba1, and Sox10 in the mPFC slices of Astrocytic ALKBH5 cKO mice and littermate controls. Confocal imaging and quantification analysis showed that the ALKBH5 positive neurons, oligodendrocytes, and microglias were not significantly changed in the astrocytic ALKBH5 cKO mice, suggesting that selective knockout of ALKBH5 in astrocytes (Supplementary Fig. 5a–c).

For behavioral studies, we first investigated the behavioral performance of Astrocyte cKO mice and littermate controls. In the force swimming test (FST), the duration of immobility was decreased in Astrocyte cKO mice compared with Ctrl mice (Fig. 2d). In the CSDS experiment, Astrocyte cKO and Ctrl mice showed no difference in social interaction in the baseline−pre-defeat conditions (Fig. 2e). However, after social defeat stress for 10 consecutive days, Ctrl mice spent less interaction time with aggressive CD1 target in SI test, whereas Astrocyte cKO mice prevented the development of the social avoidance (Fig. 2e and Supplementary Fig. 4i). In the sucrose preference test (SPT), Astrocyte cKO and Ctrl mice showed no difference in the sucrose preference test before CSDS. After 10 days CSDS paradigm, the sucrose preference was decreased in Ctrl mice, whereas Astrocyte cKO mice prevented the development of anhedonia (Fig. 2f). After mice were exposed to anxiety-related behavioral tests, Astrocyte cKO mice spent less time in the open arms of elevated plus maze (EPM) (Fig. 2g), but no behavioral differences were observed in the light-dark box (LD) (Supplementary Fig. 4k). Astrocyte cKO mice showed a slight decrease in locomotor performance in the open field test (OFT) (Supplementary Fig. 4g, j), but no differences were observed in motor coordination in the rotarod test (Supplementary Fig. 4h).

Then, we investigated the behavioral performance of Neuron cKO and EC cKO mice (Fig. 2a). No obvious behavioral changes were observed between the Neuron cKO and Ctrl mice in FST, SI test before and after CSDS and OFT (Fig. 2j, k and Supplementary Figs. 6e–h). Neuron cKO mice spent less time in the open arm in the EPM (Fig. 2l) and more time in the dark box in the LD (Supplementary Fig. 6i–k). For EC cKO mice and littermate controls, no obvious changes were observed in the FST, SI test before and after CSDS, OFT, LD, and EPM (Fig. 2o–q and Supplementary Fig. 7b–h). Together, these data indicate that selective knockout of ALKBH5 in adult astrocytes produces antidepressant-like behaviors, not in neurons or endothelial cells.

## Astrocytic ALKBH5 in the mPFC bidirectionally mediates depression-related behaviors

To determine the brain region-specific effect of ALKBH5 on behaviors, we specifically knocked out ALKBH5 in the mPFC of adult *Alkbh5*$^{loxP/loxP}$ mice by using bilateral injection of AAV-gfaABC1D-GFP-iCre or AAV-gfaABC1D-eGFP (Fig. 3a). Confocal imaging and Western blotting analysis showed that viral transfection resulted in reduced ALKBH5 expression in the mPFC astrocytes (Fig. 3b, c). To address the specificity of AAV-gfaABC1D-eGFP-iCre expression, we immunostained S100β of the mPFC slice from *Alkbh5*$^{loxP/loxP}$ mice injected with GFAP-icre virus. Confocal imaging and quantification analysis showed that mostly eGFP-positive cells co-stained with S100β in the slices (Fig. 3b and Supplementary Fig. 8a). We next examined behavioral performance and found that the mice infected with the AAV-gfaABC1D-GFP-iCre virus (GFAP$^{△Alkbh5}$) exhibited antidepressant-like behaviors, including reduced immobility time in the FST (Fig. 3d) and prevented the development of the social avoidance in SI after CSDS paradigm (Fig. 3e and Supplementary Fig. 8e). No differences were observed between GFAP$^{△Alkbh5}$ and Ctrl mice in EPM, OFT and LD (Fig. 3f–h and Supplementary Fig. 8b–d).

We then tested whether overexpression of ALKBH5 in the mPFC astrocytes (OE-Alkbh5) was sufficient to induce depressive-like behaviors by using a viral expression approach (Fig. 3i). Confocal imaging and Western blotting analysis confirmed the upregulation of ALKBH5 in the mPFC astrocytes (Fig. 3j, k). In behavioral tests, a significant increase in immobility time in FST was observed in OE-Alkbh5 mice (Fig. 3l). After that, we employed a 3-day subthreshold social defeat stress (SSDS) paradigm. Before SSDS, OE-Alkbh5, and Ctrl mice showed no difference in social interaction, but after SSDS, OE-Alkbh5 mice spent statistically less time in the interaction zone in the SI test compared to Ctrl mice (Fig. 3m and Supplementary Fig. 8j). No differences were observed between OE-Alkbh5 and Ctrl mice in EPM, OFT and LD (Fig. 3n–p and Supplementary Fig. 8f–i).

To determine whether restoration of astrocytic ALKBH5 is sufficient to reverse the antidepressant-like behaviors induced by astrocytic ALKBH5 knockout, we injected AAV-DIO-ALKBH5 or Ctrl virus into the mPFC of Astrocyte cKO mice (Fig. 3i). As expected, the Astrocyte cKO mice overexpressed ALKBH5 in mPFC showed the increase of immobility in FST (Fig. 3q) and spent less time in SI test compared with Astrocyte cKO mice infected with Ctrl virus (Fig. 3r and Supplementary Fig. 8n). No differences were observed in EPM, OFT and LD (Fig. 3s–u and Supplementary Fig. 8k–m). These findings suggest that the gain or loss of astrocytic ALKBH5 in the mPFC bidirectionally modulates depression-related behaviors.

Previous studies show that Fgfr3 is also expressed in the interneurons of the olfactory bulb[24]. To rule out the effect of the *Fgfr3-iCre* mouse line in driving recombination in the olfactory bulb interneuron, we injected AAV2/9-Syn-eGFP-WPRE-pA virus into the olfactory bulb of *Alkbh5*$^{loxp/loxp}$ mice (iCre) (Supplementary Fig. 9a). Western blotting analysis showed that viral transfection resulted in reduced ALKBH5 expression in olfactory bulb (Supplementary Fig. 9b). Behavioral tests were performed, and results showed that no significant difference was observed in FST (Supplementary Fig. 9c). The iCre mice showed a decrease in locomotor performance in OFT (Supplementary Fig. 9d–g). And the iCre mice spent less time in the open arms of EPM and less time in LD (Supplementary Fig. 9h–k). These results exclude the influence of ALKBH5 in the olfactory bulb interneuron in depression-related behaviors.

## ALKBH5 alters the m6A methylation and epitranscriptome of mPFC astrocytes

To investigate the mechanism by which astrocytic ALKBH5 regulates depression-related behavior, we collected mPFC tissues and performed m6A epitranscriptomic analysis by MeRIP-Seq (Supplementary Fig. 10a, b). Consistent with previous studies[25], m6A modification was increased mainly in the 3′ UTR, rather than in the 5′ UTR or coding

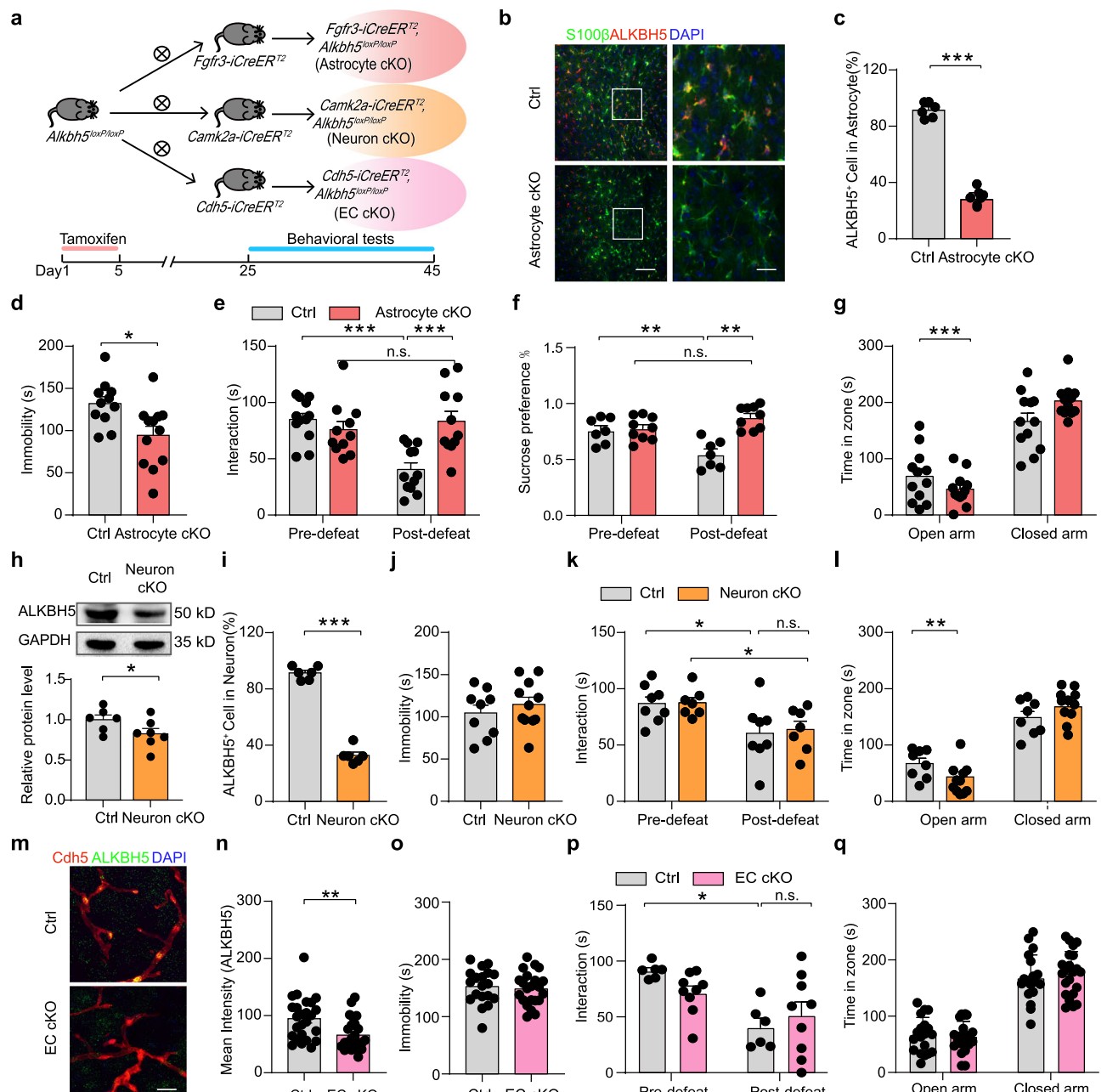

**Fig. 2 | Astrocyte-specific knockout of ALKBH5 induces antidepressant-like behaviors. a** Genetic crosses used to delete the ALKBH5 of astrocytes, neurons, and endothelial cells from the whole brain and a schematic representation of the behavioral tests. **b**, **c** Representative images (red, ALKBH5; green, S100β) (**b**) and quantification of the ALKBH5 positive astrocytes (**c**) of Astrocyte cKO and Ctrl mice, Scale bar = 20 μm. (*n* = 6 mice per group). **d**, **j**, **o** Immobility time in the FST. (**d** Astrocyte cKO, *n* = 12; Ctrl, *n* = 11 mice; **j** Neuron cKO, *n* = 11; Ctrl, *n* = 9 mice; **o** EC cKO, *n* = 22; Ctrl, *n* = 19 mice). **e**, **k**, **p** Time spent in the interaction zone before and after CSDS. (**e** Astrocyte cKO, *n* = 11; Ctrl, *n* = 12 mice; **k** Neuron cKO, *n* = 7; Ctrl, *n* = 8 mice; **p** EC cKO, *n* = 9; Ctrl, *n* = 7 mice). **f** Sucrose preference for Astrocyte cKO and Ctrl mice in the SPT. (Astrocyte cKO, *n* = 7; Ctrl, *n* = 9 mice). **g**, **l**, **q** Time spent in the

open arms and closed arms in the EPM. (**g** Astrocyte cKO, *n* = 12; Ctrl, *n* = 12 mice; **l** Neuron cKO, *n* = 8; Ctrl, *n* = 11 mice; **q** EC cKO, *n* = 22; Ctrl, *n* = 19 mice). **h** Western blotting analysis of the ALKBH5 in the mPFC of Neuron cKO mice. (Neuron cKO, *n* = 7; Ctrl, *n* = 6 mice). **i** Quantification of the ALKBH5 positive neurons of Neuron cKO and Ctrl mice, Scale bar = 20 μm. (*n* = 6 mice per group). **m**, **n** Representative images and quantification of the ALKBH5 positive endothelial cells (red, Cdh5; green, ALKBH5) of EC cKO and Ctrl mice, Scale bar = 20 μm. (*n* = 25 slices from 4 mice per group). All data were presented as the mean ± SEM. Two-sided unpaired *t*-test (**c**, **d**, **g**–**j**, **l**, **n**, **o**, **q**) or Two-way ANOVA with Bonferroni's multiple comparisons test (**e**, **f**, **k**, **p**). **p* < 0.05; ***p* < 0.01; ****p* < 0.001; n.s. no significance. Source data are provided as a Source Data file. See Supplementary Data 4 for statistical details.

region (Fig. 4a, b), and 'GGAC', the most frequent m6A consensus motifs, were contained in the enriched motifs in all samples (Fig. 4c). Next, we analyzed the differentially methylated peaks (DMPs) between Astrocyte cKO and littermate controls (Fig. 4d). Most DMPs were distributed in exons and the 3' UTR, and the overall differentially expressed genes (DEGs) profiles of the Astrocyte cKO and Ctrl groups were obtained with a clear separation after hierarchical cluster analysis

(Fig. 4e). Furthermore, by way of pathway analyses with Gene Ontology (GO) and Kyoto Encyclopedia of Genes and Genomes (KEGG) the Glutamatergic synapse pathway was significantly enriched in the dataset (Fig. 4f, g). The pathway includes the glutamate transporter SLC1A2 (also known as GLT-1 or EAAT2), a glutamate transporter primarily expressed on astrocytes that scavenges glutamate from excitatory synapses and prevents glutamate excitotoxicity[26,27]. The m6A

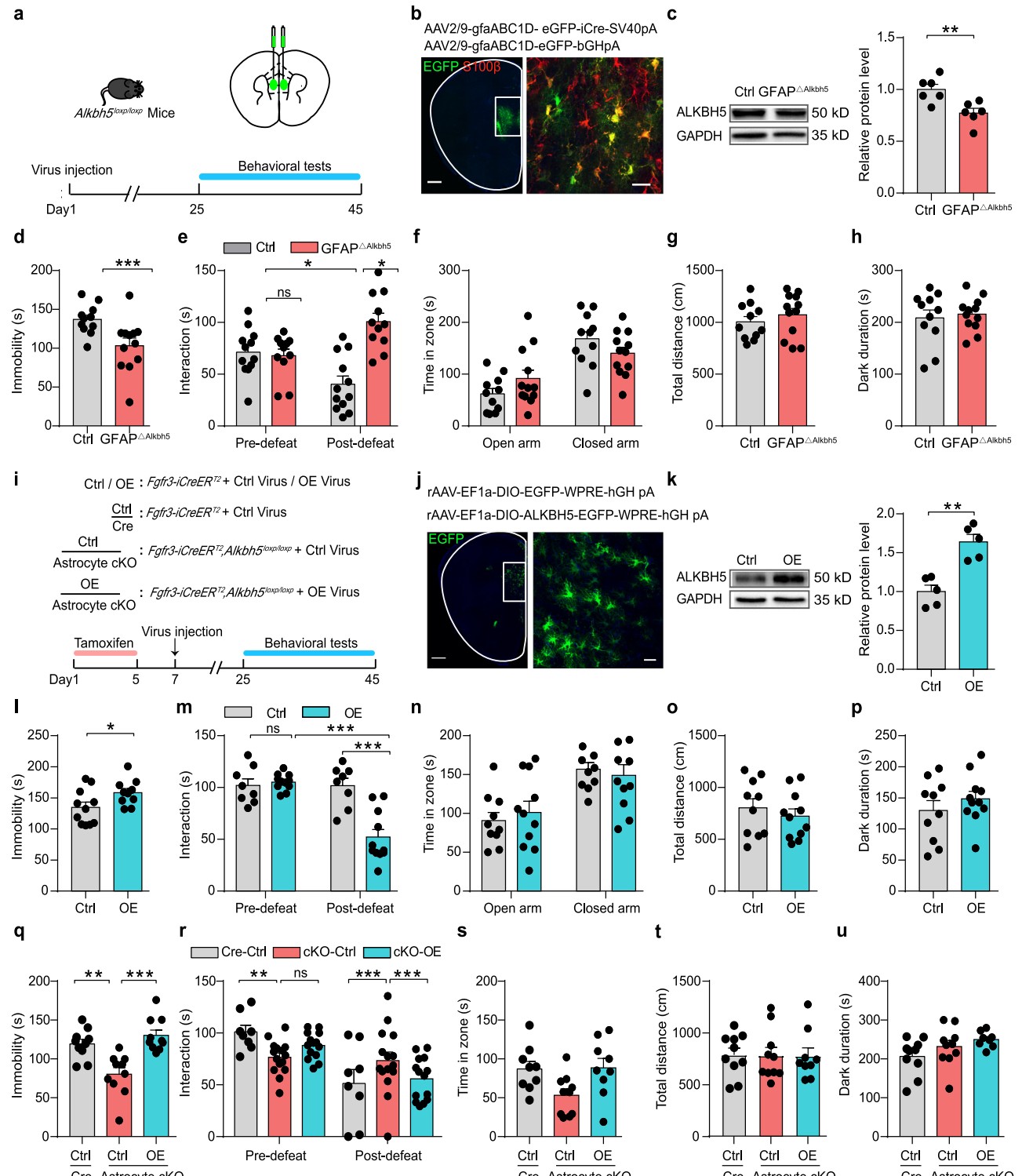

**Fig. 3 | Astrocyte-specific gain and loss of ALKBH5 in the mPFC mediates depression-related behaviors. a, i** Schematic of experimental paradigm. **b** Representative images of AAV-gfaABC1D-GFP-iCre expression in the mPFC of *Alkbh5*[loxP/loxP] mice (green, EGFP; red, S100β). Scale bars = 500 µm (left), 20 µm (right). (*n* = 6 replicates from three mice). **c** Western blotting analysis of ALKBH5 in the mPFC of GFAP[△Alkbh5] and Ctrl mice. (*n* = 6 mice per group). **d–h** Statistics analysis of the GFAP[△Alkbh5] and Ctrl mice in FST (**d**), SI test (**e**), EPM (**f**), OFT (**g**), and LD (**h**). (**d**, **f–h** Ctrl *n* = 11; GFAP[△Alkbh5] *n* = 12 mice; **e** Ctrl *n* = 12; GFAP[△Alkbh5], *n* = 11 mice). **j** Representative images of the AAV-DIO-ALKBH5 expression in the mPFC of *Fgfr3-iCreER*[T2] mice. Scale bars = 500 µm (left), 20 µm (right). (*n* = 6 replicates from three mice). **k** Western blotting analysis of the ALKBH5 in the mPFC of ALKBH5 OE mice. (*n* = 5 mice per group). **l–p** Statistics analysis of OE-Alkbh5 and Ctrl mice in FST

(**l** OE, *n* = 10; Ctrl, *n* = 11 mice); SI test (**m** OE, *n* = 11; Ctrl, *n* = 8 mice); EPM (**n**), OFT (**o**), and LD (**p**), (**n–p**, OE, *n* = 11; Ctrl; *n* = 10 mice). **q–u** Statistical analysis of Astrocyte cKO mice that injected with AAV-DIO-ALKBH5 or AAV-DIO-eGFP in FST (**q** Cre-Ctrl, *n* = 10; cKO-Ctrl, *n* = 10; cKO-OE, *n* = 10 mice); SI test (**r** Cre-Ctrl, *n* = 8; cKO-Ctrl, *n* = 15; cKO-OE, *n* = 14 mice); EPM (**s** Cre-Ctrl, *n* = 9; cKO-Ctrl, *n* = 10; cKO-OE, *n* = 9 mice); OFT (**t** Cre-Ctrl, *n* = 10; cKO-Ctrl, *n* = 10; cKO-OE, *n* = 9 mice); LD (**u** Cre-Ctrl, *n* = 10; cKO-Ctrl, *n* = 10; cKO-OE, *n* = 8). All data were presented with the mean ± SEM. Two-sided unpaired *t*-test (**c**, **d**, **f–h**, **k**, **l**, **n–p**) or one-way ANOVA followed by Bonferroni's test for multiple comparisons (**q**, **s–u**), and two-way ANOVA with Bonferroni's multiple comparisons test (**e**, **m**, **r**). *p < 0.05; **p < 0.01; ***p < 0.001; n.s. no significance. Source data are provided as a Source Data file. See Supplementary Data 4 for statistical details.

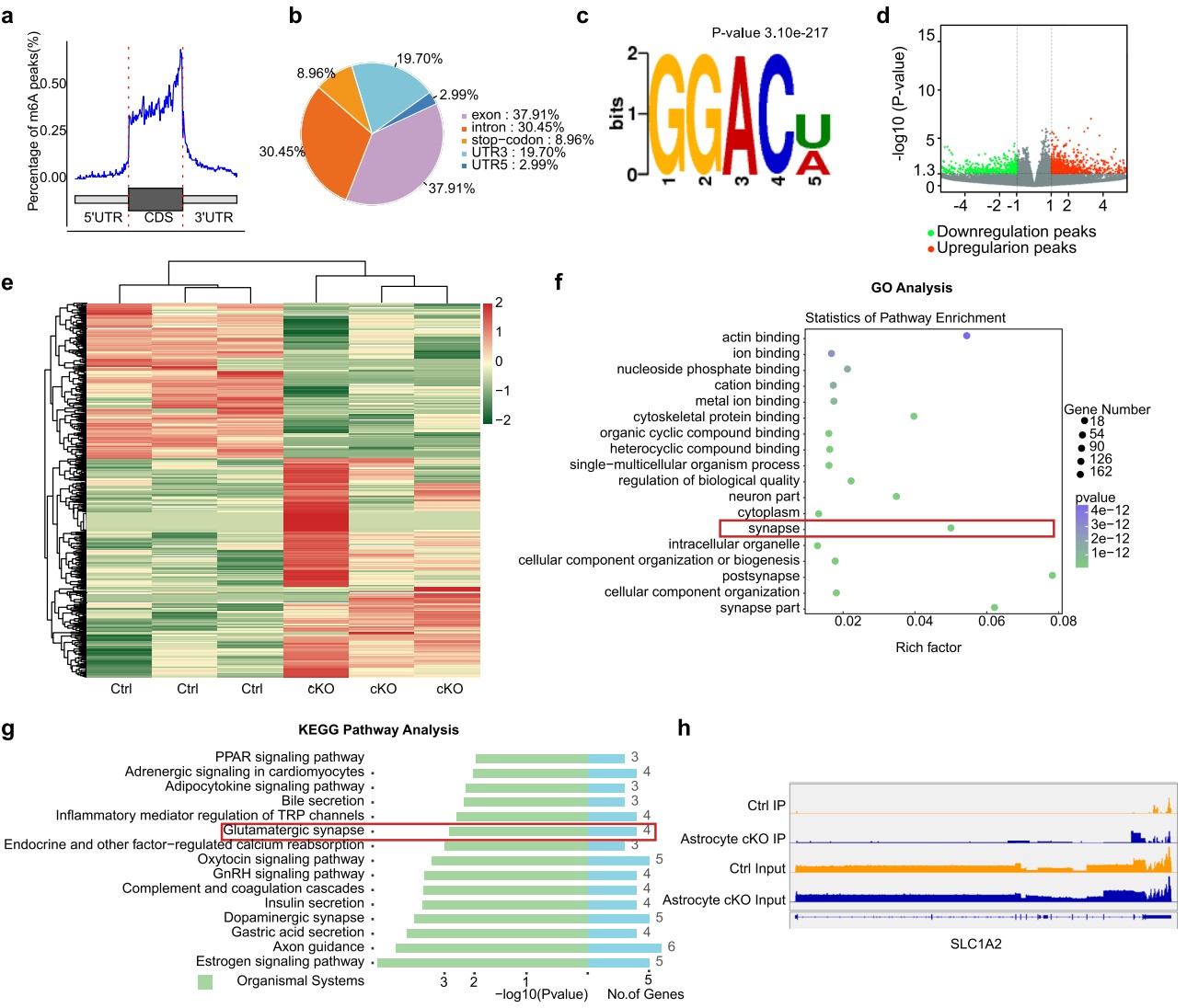

**Fig. 4 | Astrocytic ALKBH5 deletion increases m6A levels and alters mPFC epitranscriptome. a** Transcriptome-wide distribution of m6A peaks of control samples. **b** Number and percent of m6A peaks. **c** Consensus motif of m6A sites identified in m6A peaks of control samples. **d** Volcano diagram showing overlap of genes with significant changes in mRNA expression in the Astrocyte cKO mice and Ctrl. **e** Clustering and heatmaps showing mRNA levels in the mPFC Astrocyte cKO and Ctrl mice. Relative mRNA expression level is represented in color. Only m6A genes with enrichment fold >1 in controls and significant differences in mRNA expression between the experimental and control groups. **f** GO analysis of mRNAs with reads per kilobase of transcript, per million mapped reads (RPKM) ratio <1 and $p < 0.05$. **g** KEGG analysis with FC >1 and $p < 0.05$. **h** m6A modification on SLC1A2 gene. Two-tailed Fisher's exact test and FDR corrected $p < 0.05$ were considered significant (**d**–**g**). Two-sided Fisher's exact test with adjustments for multiple testing (**c**). See Supplementary Data 4 for statistical details.

modification of GLT-1 was significantly increased in the mPFC of Astrocyte cKO mice compared to the Ctrl group, according to quantitative analysis of consensus peaks (Fig. 4h). These findings suggest that astrocytic ALKBH5 may be involved in glutamatergic synapses via the m6A modification of GLT−1.

## Astrocytic ALKBH5 regulates glutamate levels via m6A methylation modification of GLT−1

To investigate whether the GLT-1 m6A sites regulate the expression of GLT−1, we cloned two mutants at the m6A modification site of the 3'UTR of GLT−1 (Fig. 5a) into the psiCHECK-2 double luciferase expression vector. The luciferase activity was reduced in cells transfected with the SLC1A2 mutant vector compared with the empty vector control, suggesting that GLT-1 3' UTR m6A site regulate GLT−1 expression (Fig. 5b). Then, we used a single-base editing to construct dCas9-mCherry and GLT-1-targeting-sgRNA-puro-eGFP and then transfected them into mixed primary cultured astrocytes to alter the 3' UTR region m6A modification site of GLT-1 (Fig. 5c). Western blotting

analysis showed that GLT-1 protein levels were significantly decreased in the astrocytes with GLT-1-targeting-sgRNA-puro-eGFP compared with the Ctrl plasmid (Fig. 5d). Then we determined the glutamate uptake capacity of astrocytes by measuring glutamate levels in the medium after 4 h of L-glutamate (200 M) treatment and found that the concentration of glutamate in the sgRNA group was almost 40% higher than that in the Ctrl group, indicating 201 mutant GLT-1 disrupted the glutamate uptake ability of astrocytes (Fig. 5e).

To examine the contribution of astrocytic ALKBH5 in the m6A modification of GLT-1, we then examined the expression and uptake ability of GLT-1 in ALKBH5 cKO astrocytes in vitro. Western blotting analysis showed a decrease in ALKBH5 and an increase in GLT-1 in the primary cultured astrocytes from *Alkbh5 loxP/loxP* mouse infected with pLenti-GFAP-Cre virus (Fig. 5f). Meanwhile, the level of uptaken glutamate was increased in ALKBH5 cKO astrocytes compared with the Ctrl (Fig. 5g). Furthermore, we examined GLT-1 expression and extracellular glutamate dynamics in the mPFC of Astrocyte cKO mice. Western blotting analysis revealed increased GLT-1 in the

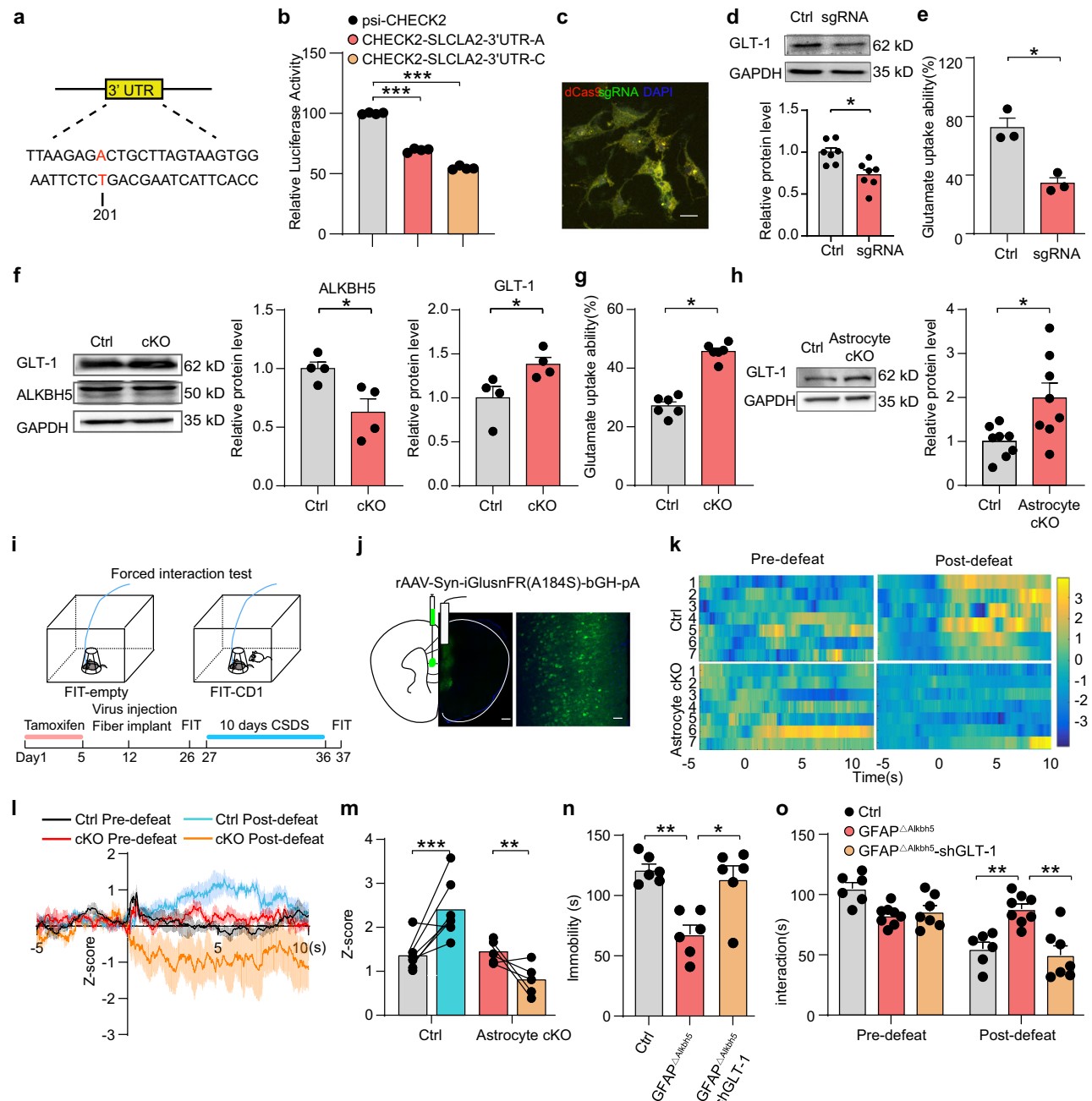

**Fig. 5 | Astrocytic ALKBH5 modulates glutamate levels through methylation modification of GLT-1. a** Targets to specifically alter m6A modification sites of GLT-1. **b** Dual-luciferase reporter constructs with the psiCHECK-2 vector contain Renilla luciferase and firefly luciferase driven by two different promoters. The effects of GLT-1 m6A were examined by transfection of a dual-luciferase reporter. (*n* = 4 biological replicates). **c** For single-base editing to construct and cotransfect dCas9-mCherry and sgRNA-puro-eGFP into mixed primary cultured astrocytes and representative images. Scale bars = 20 μm (right). (*n* = 6 replicates from three wells). **d** Western blotting analysis of ALKBH5 of sgRNA and Ctrl group. (*n* = 7 wells per group). **e** Glutamate uptake ability in the medium of astrocytes transfected with sgRNA and Ctrl plasmids. (*n* = 3 wells per group). **f** Western blotting analysis of ALKBH5 and GLT-1 in *Alkbh5loxP/loxP* astrocytes infected pLent-GFAP-Cre virus (cKO) and Ctrl virus. (*n* = 4 wells per group). **g** Glutamate uptake ability in the medium of cKO and Ctrl astrocytes. (*n* = 6 mice per group). **h** Western blotting analysis of GLT-1 in the mPFC of astrocyte cKO and Ctrl mice. (*n* = 8 mice per group). **i, j** Schematic

illustrating fiber placement and representative images of AAV-Syn-iGluSnFR(A184S) expression. Scale bars = 500 μm (left), 20 μm (right). (*n* = 6 replicates from three mice). **k** Representative heatmaps of z-scores changes over all trials from single mice. Each row plots one trial, and a total of four trials are illustrated. **l, m** Time course of average iGluSnFR transient z-scores event locked to social interaction (**l**) and peak z-score (**m**) during social interaction. (Astrocyte cKO, *n* = 5; Ctrl, *n* = 7 mice). **n, o** Statistical analysis of GFAP△Alkbh5 mice injected with AAV-shRNA(GLT-1) or AAV-mCherry in FST (**n** *n* = 6 mice per group); SI test (**o** Ctrl, *n* = 6; GFAP△Alkbh5, *n* = 8; GFAP△Alkbh5-shGLT-1, *n* = 7 mice). All data were presented as the mean ± SEM. Two-sided unpaired *t*-test (**d–h**) or one-way ANOVA followed by Bonferroni's test for multiple comparisons (**b, n**), or two-way ANOVA (**m, o**) with Bonferroni's multiple comparisons test. **p* < 0.05; ***p* < 0.001; ****p* < 0.001; n.s. no significance. Source data are provided as a Source Data file. See Supplementary Data 4 for statistical details.

mPFC of Astrocyte cKO mice compared with the littermate control (Fig. 5h).

To detect extracellular glutamate dynamics in vivo, we employed a glutamate sensor (iGluSnFR, A184S) containing a high-affinity SF-iGluSnFR variant that enhances the detection of stimulus-evoked glutamate release[28]. After the Astrocyte cKO and Ctrl mice were further treated through injections of AAV-Syn-iGluSnFR(A184S) in the mPFC, fiberoptic implants were placed above the infected cells (Fig. 5i). Two weeks after the viral expression, photometry recordings were performed in chronically stressed mice with a forced interaction test (FIT) assay (Fig. 5j). Before CSDS paradigm, the heatmaps and maximum fluorescence change (Z-score) showed that similar amounts of glutamate was released in astrocytic cKO and Ctrl mice when interacting with CD1 in FIT. After 10 days of CSDS, the amount of glutamate was increased in the Ctrl mice to a large extent as well during CD1 contact, but the extent of glutamate increase in the astrocytic cKO mice was notably reduced (Fig. 5k–m).

Furthermore, we specifically knocked down astrocytic GLT-1 by using AAV- gfaABC1D-GLT-1-shRNA. GLT-1-shRNA or Ctrl virus was injected into the mPFC of GFAP$^{\triangle Alkbh5}$ mice (Supplementary Fig. 11a). Western blotting analysis showed that GLT-1-shRNA reversed the elevation of GLT-1 level in the mPFC of GFAP$^{\triangle Alkbh5}$ mice (Supplementary Fig. 11b). The GFAP$^{\triangle Alkbh5}$ mice infected with GLT-1-shRNA showed an increase in immobility in the FST (Fig. 5m) and spent less time in the SI test after CSDS (Fig. 5n and Supplementary Fig. 11g), suggesting that GLT-1-shRNA reversed the antidepressant-like effects of GFAP$^{\triangle Alkbh5}$ mice. No differences were observed in EPM, OFT, and LD (Supplementary Fig. 11c–f). These results suggest that astrocytic ALKBH5 mediates depression-related behaviors via GLT-1 m6A methylation.

### Knockout of ALKBH5 in astrocytes protects glutamatergic synaptic transmission under stress

To investigate whether ALKBH5 in astrocytes resists stress by affecting synaptic transmission, we performed the whole-cell patch clamp to record the action potentials from layer IV-V pyramidal neurons in the astrocytic ALKBH5 cKO and Ctrl mice. We found no difference in the amplitude and number of action potentials in Ctrl and astrocytic ALKBH5 cKO mice with the presence or absence of stress (Fig. 6a–c). Next, we recorded spontaneous excitatory postsynaptic currents (sEPSCs) in the mPFC. After 10 days CSDS, the frequency of sEPSCs in the Ctrl mice was significantly decreased, but remained unchanged in the astrocytic ALKBH5 cKO mice in response to stress (Fig. 6d, e and Supplementary Fig. 12a, b), and meanwhile, the amplitude of sEPSCs was observed with no difference in the mice of two groups (Fig. 6f).

Then, miniature excitatory postsynaptic currents (mEPSCs) in the pyramidal neurons were determined to investigate the effects of basal glutamatergic synaptic transmission in the mPFC of Astrocyte ALKBH5 cKO and Ctrl mice. The results showed that the frequency and amplitude of mEPSC were slightly decreased after conditional knockdown of ALKBH5 in the astrocytes under basal conditions (Fig. 6g, i). After CSDS, the frequency of mEPSCs was significantly decreased in the Ctrl mice, but slightly increased in the astrocytic ALKBH5 cKO even under stress (Fig. 6g, h and Supplementary Fig. 12c, d), and meanwhile, the amplitude of mEPSCs was not affected by stress (Fig. 6g, i and Supplementary Fig. 12c, d). Furthermore, we analyzed the decay times of individual mEPSCs. Under basal conditions, we observed that the decay time was decreased after the conditional knockdown of ALKBH5 in the astrocytes (Fig. 6j, k). After CSDS, the decay time of mEPSCs was elevated in the Ctrl mice, indicating longer glutamate clearance from synapses (Fig. 6j, k), and it was slightly increased in the astrocytic ALKBH5 cKO even under stress (Fig. 6j, k). Next, mEPSCs in the pyramidal neurons of GFAP$^{\triangle Alkbh5}$ mice infected with GLT-1-shRNA were also recorded, and the results showed that both the frequency and amplitude of mEPSCs were slightly decreased after conditional

knockdown of ALKBH5 in astrocytes, which was reversed by GLT-1-shRNA (Supplementary Fig. 12e, f). These findings implied that the knockout of astrocytic ALKBH5 prevented attenuation of glutamatergic synaptic transmission during chronic stress.

### Astrocytic ALKBH5 modulates neuronal morphology and Ca$^{2+}$ activity under social stress

As excitotoxicity induced by chronic stress or abnormally high levels of extracellular glutamate is reported to cause dendritic atrophy and spinal loss, leading to neuronal dysfunction[29–31], we used sparingly labeled layer II/III pyramidal neurons in the mPFC of Astrocyte cKO and Ctrl mice with a viral cocktail (1:1) of AAV-CaMKII-FLP and AAV-nEf1-FDIO-EYFP to evaluate neuronal morphological alterations in response to chronic stress. Morphological analysis showed that both Astrocyte cKO and Ctrl mice exhibited no significant differences in total dendritic length, dendritic complexity, and spine density of pyramidal neurons before the CSDS experiment, but all of them were significantly decreased in Ctrl mice after the CSDS experiment (Fig. 7a–e). Notably, the total dendritic length, dendritic complexity, and spine density in the Astrocyte cKO mice were relatively intact after CSDS (Fig. 7d, e). These findings indicated that loss of astrocyte ALKBH5 function reduced dendritic and spinal loss of pyramidal neurons in response to social stress.

Next, to determine the neuronal activity under stress, we employed in vivo fiber photometry and the Ca$^{2+}$ indicator, GCaMP6s. In order to do this, we injected the AAV-Syn-GCaMP6s virus into the mPFC, and implanted optical fibers above the infected cells (Fig. 7f, g). After two weeks, photometry recordings were performed in chronically stressed mice with the FIT (Fig. 7f). Before the CSDS experiment, heatmaps and maximum fluorescence change (Z-score) showed that both Astrocyte cKO and Ctrl mice demonstrated similar stimulus-evoked intracellular Ca$^{2+}$ elevations while interacting with a CD1 aggressor in FIT (Fig. 7h, i). After 10 days of CSDS, GCaMP6s fluorescence was substantially reduced in the Ctrl mice during the contact period, and the Ca$^{2+}$ activity in the Astrocyte cKO mice notably showed no significant change during CD1 exposure (Fig. 7i–k), indicating that astrocyte ALKBH5 deficiency maintained Ca$^{2+}$ activity in the mPFC pyramidal neurons under social stress.

To explore astrocytic ALKBH5 modulate depression-related behaviors via glutamatergic neuronal activity, we bidirectionally injected the AAV-CamKIIα-hM4D(Gi)-mCherry virus into the mPFC of Astrocyte cKO mice to specifically manipulate the activity of glutamatergic neurons. Confocal images showed that hM4Di was expressed in glutamatergic neurons (Supplementary Fig. 13a). Electrophysiologically, the CNO (1 μm) significantly inhibited the firing of hM4Di-expressing glutamatergic neurons (Supplementary Fig. 13b). Behaviorally, intraperitoneal injection of CNO dramatically in Astrocyte cKO mice reversed the decrease of immobility of Astrocyte cKO mice in FST (Supplementary Fig. 13c). No differences were observed in LD and EPM (Supplementary Fig. 13d, e). These results suggest that the m6A modification of GLT-1 plays a critical role in regulating depression-related behaviors.

### SAMe induces antidepressant-like effects through m6A methylation in adult C57BL/6J mice

S-adenosylmethionine (SAMe), a universal methyl donor for most of the methylation reactions, is used to sustain RNA methylation (including m6A modification) via RNA methyltransferases and regulate RNA splicing, processing, translation, and decay (Fig. 8a)[4,32]. To investigate the role of the SAMe in m6A modification dysfunction in depression, we first examined plasma SAMe levels in MDD patients. The results showed that the plasma SAMe levels were lower in the MDD patients compared with healthy controls (Fig. 8b). The plasma SAMe levels were also decreased in the LPS-induced depressed mice compared to the Ctrl mice (Fig. 8c). In addition, the SAMe concentrations in

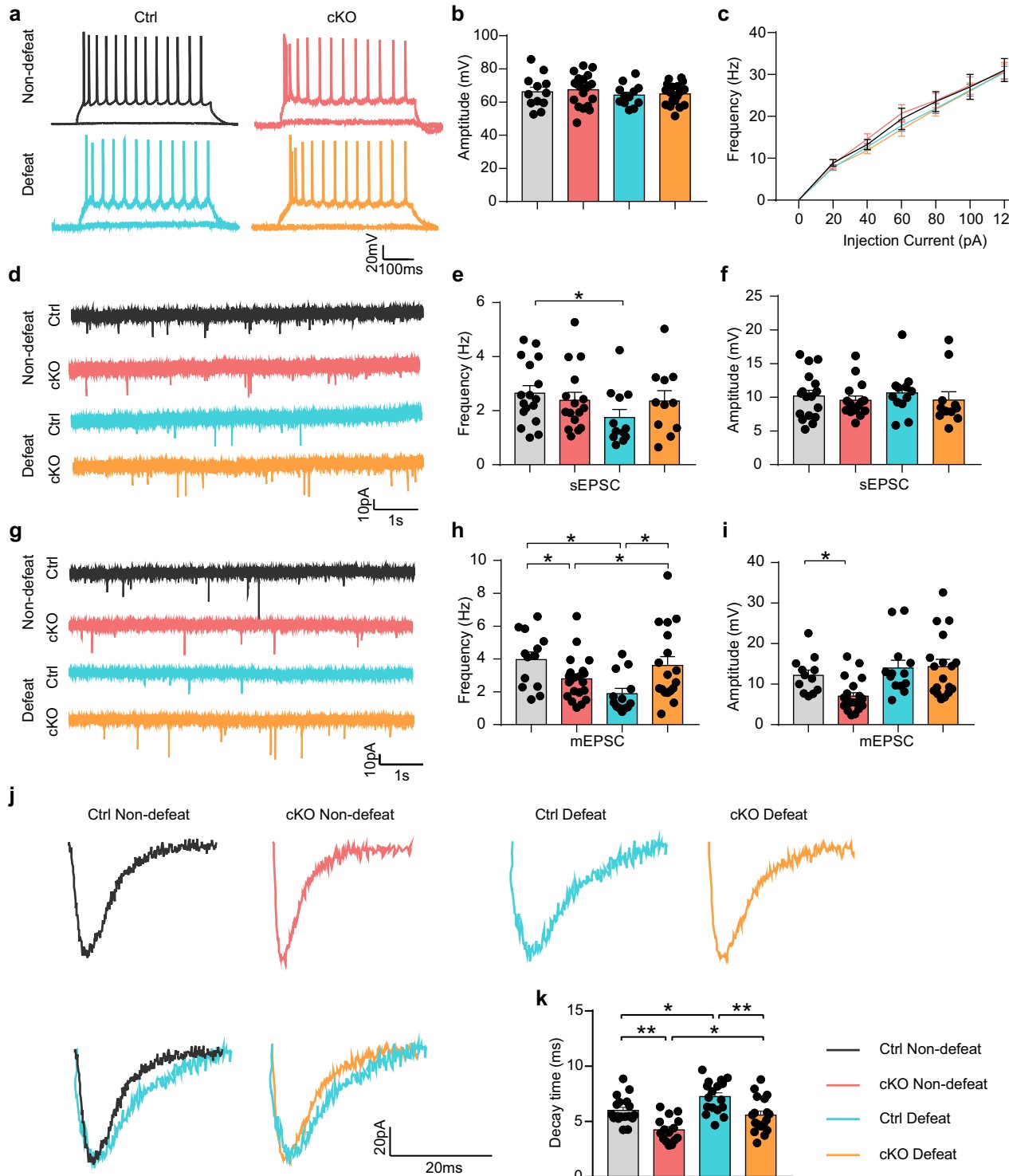

**Fig. 6 | Knockout of astrocytic ALKBH5 prevents the disruption of glutamatergic synaptic transmission from social stress. a** Representative traces of action potentials measured by whole-cell current-clamp recordings from mPFC layer IV-V pyramidal neurons of Ctrl and Astrocyte cKO mice with or without the CSDS experiments. Representative voltage traces in neurons showing the effects of a series of 500 ms current pulses ranging from 0 to 120 pA in 20 pA steps. **b, c** Quantification of the amplitude of action potentials currents (**b** Ctrl, $n = 12$; Astrocyte cKO, $n = 20$ cells from four individual mice) and Summarized results of firing rate under increasing step currents (**c** Ctrl, $n = 12$; Astrocyte cKO, $n = 21$ cells from four individual mice) of pyramidal neurons from Ctrl and Astrocyte cKO mice with or without the CSDS paradigm. **d** Representative traces of sEPSCs recorded from mPFC neurons from Ctrl and Astrocyte cKO mice with or without the CSDS. **e, f** Quantification of sEPSCs frequency (**e**) and amplitude (**f**). (Ctrl Non-defeat,

$n = 17$; cKO Non-defeat, $n = 16$; Ctrl Defeat, $n = 12$; cKO Defeat, $n = 11$ cells from four individual mice). **g** Representative traces of mEPSCs recorded from mPFC neurons from Ctrl and Astrocyte cKO mice with or without the CSDS. **h, i** Quantification of mEPSC frequency (**h**) and amplitudes (**i**). (Ctrl Non-defeat, $n = 13$ in (**h**), $n = 12$ in (**j**); cKO Non-defeat, $n = 21$; Ctrl Defeat, $n = 13$; cKO Defeat, $n = 18$ cells from four individual mice). **j** Representative traces of mean individual mEPSCs from Ctrl and Astrocyte cKO mice with or without the CSDS. **k** Quantification mean value of mEPSC decay time. (Ctrl Non-defeat, $n = 18$; cKO Non-defeat, $n = 15$; Ctrl Defeat, $n = 17$; cKO Defeat, $n = 18$ cells from four individual mice). All data were presented as the mean ± SEM. Two-way ANOVA with Bonferroni's multiple comparisons test (**b**, **c**, **e**, **f**, **h**, **i**, **k**). *$p < 0.05$; ***$p < 0.001$; n.s. no significance. Source data are provided as a Source Data file. See Supplementary Data 4 for statistical details.

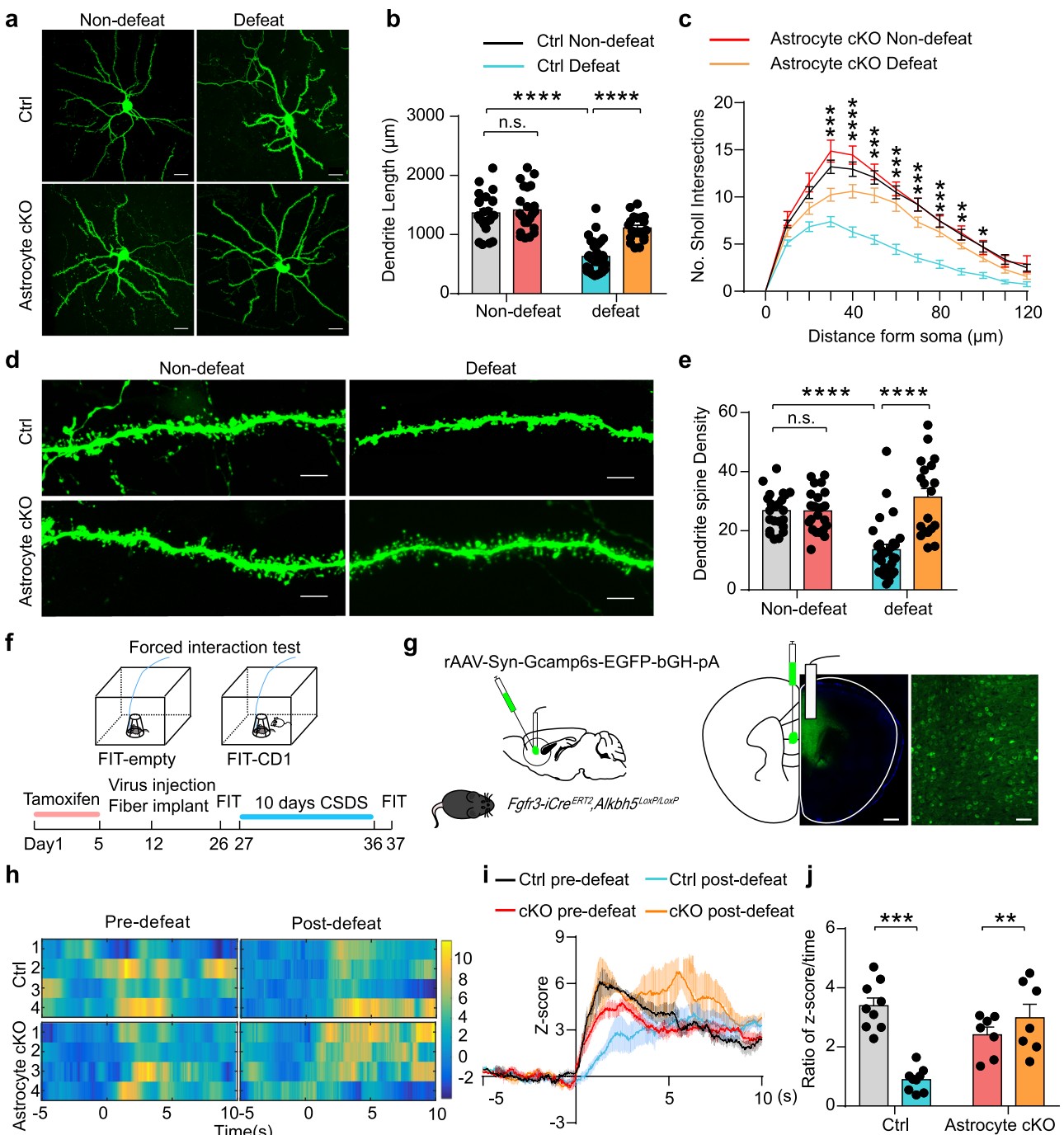

**Fig. 7 | Astrocytic ALKBH5 preserves neuronal morphology and Ca²⁺ activity under social stress. a–c** Representative confocal images showing that pyramidal neurons infected with AAV-fDIO-EGFP and AAV-FLP (**a**), quantification of total dendrite length (**b**), and Sholl analysis (**c**) in the mPFC of Astrocyte cKO or Ctrl mice after CSDS paradigm. Scale bar = 20 μm. (Ctrl Non-defeat, $n = 24$; cKO Non-defeat, $n = 22$; Ctrl Defeat, $n = 31$; cKO Defeat, $n = 20$ cells from six individual mice). **d, e** Representative images of dendritic segments (**d**), quantification of total spine density (**e**). Scale bar = 5 μm. (Ctrl Non-defeat, $n = 23$; cKO Non-defeat, $n = 22$; Ctrl Defeat, $n = 29$; cKO Defeat, $n = 19$ cells from six individual mice). **f** Schematic illustrating fiber photometry. **g** Schematic illustrating fiber placement and

representative images of GCamp6s expression. Scale bars = 500 μm (left), 20 μm (right). ($n = 6$ replicates from three mice). **h** Representative heatmaps of GCaMP6s transient z-scores event locked to social interaction. Each row plots one trial, and a total of four trials are illustrated. **i, j** Average and peak z-score changes during social interaction. (Astrocyte cKO, $n = 7$; Ctrl, $n = 9$ mice). All data were presented as the mean ± SEM. Two-way ANOVA with Bonferroni's multiple comparisons test (**b**, **c**, **e**, **i**, **j**). *$p < 0.05$; **$p < 0.01$; ***$p < 0.001$; ****$p < 0.0001$ and n.s. no significance. Source data are provided as a Source Data file. See Supplementary Data 4 for statistical details.

the mPFC were significantly decreased in both the LPS-induced depression model and the CSDS paradigm (Fig. 8d, e). Furthermore, we found that total m6A level and GLT-1 m6A level in the adult C57BL/6 J mice was increased after administration of SAMe (300 μg mL⁻¹, 7 d, i.p.) (Fig. 8f, g).

To test whether SAMe could induce an antidepressant-like effect, we employed the FST. It was found that the total duration of immobility in the adult C57BL/6J mice was decreased after administration of SAMe after 1 week or 3 weeks (300 μg mL⁻¹, 7 d, i.p.) (Fig. 8h, i). For CSDS experiments, the interaction time was increased in the SI test

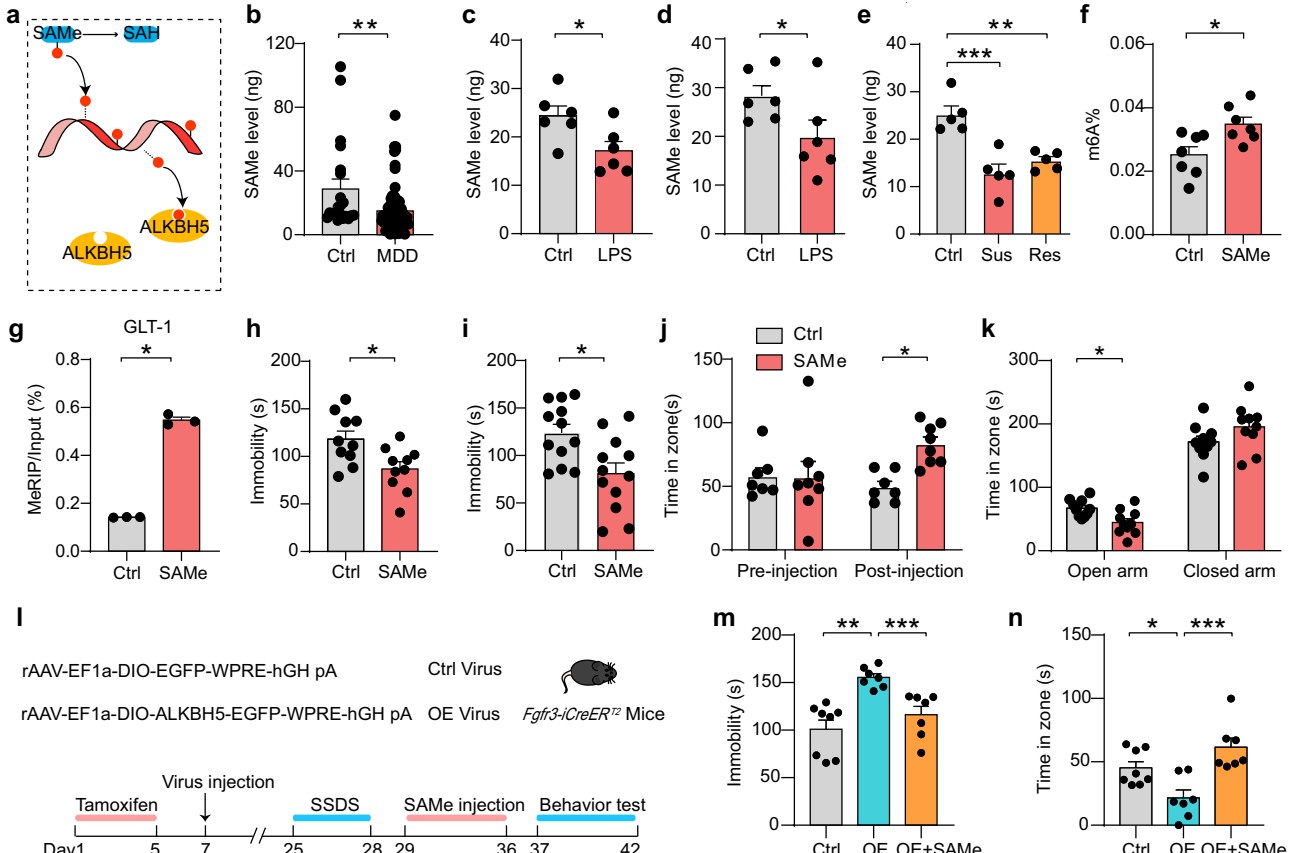

**Fig. 8 | SAMe produces antidepressant-like behaviors. a** Model of the SAMe function in m6A modification. **b** ELISA analysis of SAMe levels in the plasma of MDD patients and healthy individuals. (Ctrl, $n = 21$; MDD, $n = 69$ samples). **c** SAMe levels in the plasma of the mice with LPS-induced depression. ($n = 6$ mice per group). **d, e** SAMe levels in the mPFC of mice with LPS-induced depression (**d** $n = 6$ mice per group) and Sus, Res, and Ctrl mice after CSDS (**e** $n = 5$ mice per group). **f** m6A levels in the plasma of C57BL/6J mice treated with SAMe (300 μg mL⁻¹) or vehicle. ($n = 7$ mice per group). **g** MeRIP-qPCR of GLT-1 m6A levels in the mPFC of C57BL/6J mice treated with SAMe (300 μg mL⁻¹) or vehicle. ($n = 3$ mice per group). **h–k** Statistics analysis of C57BL/6J mice treated with SAMe or vehicle in FST (1 W) (**h** $n = 10$ mice per group), FST (3 W) (**i** $n = 12$ mice per group), SI test (**j** Ctrl, $n = 7$; SAMe, $n = 8$ mice), EPM (**k** $n = 10$ mice per group). **l** Schematic of the experimental paradigm. **m, n** Statistics analysis of OE-Alkbh5 and Ctrl mice treated with SAMe (300 μg mL⁻¹) or vehicle in FST (**m**), SI test (**n**). (Ctrl, $n = 8$; OE, $n = 7$; OE-SAMe, $n = 7$ mice). All data were presented as the mean ± SEM. Two-sided unpaired *t*-test (**b–d, f–i, k**), one-way (**e, m, n**), two-way (**j**) ANOVA with Bonferroni's multiple comparisons test. *$p < 0.05$; **$p < 0.01$; n.s. no significance. Source data are provided as a Source Data file. See Supplementary Data 4 for statistical details.

(Fig. 8j and Supplementary Fig. 14d); and less time was spent in the open arm in the EPM after SAMe treatment compared with the vehicle group (Fig. 8k). No difference in OFT and LD was observed between the SAMe and Vehicle groups (Supplementary Fig. 14a–c, e–g). In addition, we tested whether SAMe could reverse the depressive-like behaviors caused by the overexpressed ALKBH5 in the mPFC astrocytes (Fig. 8l). In FST, the immobility time caused by overexpressed ALKBH5 was reduced after SAMe treatment (Fig. 8m). After SSDS, SI test showed the OE-Alkbh5 mice spent less time on interaction zone with target while the OE-Alkbh5 mice administered with SAMe spent more time on the interaction zone with target (Fig. 8n). Taken together, these findings demonstrated that SAMe produces antidepressant-like effects through m6A methylation.

## Discussion
In this study, we reported the cell-specific and sensitive epitranscriptomic m6A of stress response in depression. Specifically, the ALKBH5 levels were increased in the blood and PFC of MDD patients as well as in the mouse models of depression; ALKBH5 in astrocytes is more sensitive to stress compared to that in neurons and endothelial cells; selective knockout of ALKBH5 in astrocytes, but not in neurons and endothelial cells, produces antidepressant-like behaviors; and under chronic stress, astrocytic ALKBH5 preserved neuronal morphology, calcium activity and glutamatergic transmission through

GLT-1 m6A modifications. Meanwhile, SAMe induced antidepressant-like behaviors by altering the global m6A modification. These results suggest that astrocytic epigenetic changes bridge the gap between the genetic and environmental factors in depression, which is indicative of a potential therapy for depression by way of targeting m6A modifications at GLT-1 in the future.

Epigenetic change is a key mechanism by which environmental stress factors contribute to the development of depressive symptoms[4]. Nevertheless, the cellular specificity and sensitivity of the environment to the epigenomics of brain transcription remains unknown. Now that most of studies have focused on the epigenetic effects of neurons, few have investigated the epigenetic effects on the perspective of astrocytic or other brain cells[9,33–35]. Astrocytes are known to play a critical role in maintaining BBB and CNS homeostasis, preventing potentially harmful signals in the blood from entering the brain[15,36]. Also, astrocytes are also considered to sense stressors from plasma more preferentially than neurons[13]. Moreover, astrocytes wrap the majority of excitatory synapses and are involved in the regulation of stress responses by way of regulating neuronal transmission[11,37]. Astrocytic epigenetic changes, including DNA, Histone, and RNA function mechanically to fine-tune multiple processes governing such fundamental cellular processes as signal transduction, transcription, and translation[38,39]. Aberrant astrocytic epigenetic changes have been identified in patients with PD, AD, ALS, and major depression[38].

However, no studies have been done on the cause-effect relationship between astrocytic epigenetics and CNS disorders like depression. In our present study on the relationship, we found that ALKBH5, an RNA demethylase of N6-methyladenosine (m6A), was increased both in the MDD patients and in the mouse models of depression (Fig. 1). Moreover, the astrocytic ALKBH5 in the mPFC regulated depression-related behaviors (Figs. 2 and 3). All findings point to the conclusion that astrocytic ALKBH5-m6A modulates depression-related behaviors. Previous studies report that ALKBH5 and FTO have a close association with MDD and are certain variations of SNP rs12936694 and SNP rs9939609, respectively, in MDD patients[40,41]. In one study, global m6A methylation is decreased in the whole blood of both human and mice after acute stress, and in another study, ALKBH5 and FTO as m6A demethylases and METTL3, METTL14, and WTAP as m6A methyltransferases are altered in the MDD patients and the depression models. In their further experiments, knockdown of FTO in adult hippocampal pyramidal neurons via siRNA and AAV-Cre virus produces depressive-like behaviors in FST, TST, and sucrose preference test and conditional deletion of METTL3 or FTO in adult hippocampus excitatory neurons of Mettl3[loxP] or FTO[loxP] mice with Nex-CreER[T2] Driver Line increases fear memory[8]. These findings emphasize the role of the m6A in the progression of depression. In our DXMS/LPS dose-dependent and time-dependent experiments, the changes in ALKBH5 protein expression in the cultured astrocytes were more sensitive than those in the neurons and endothelial cells (Fig. 1). The astrocytic ALKBH5 cKO mice in our experiments exhibited antidepressant-like behaviors in FST and CSDS (Fig. 2), and mPFC astrocytic ALKBH5 bidirectionally regulated depression-related behaviors (Fig. 3). In contrast, no significant changes in the depression-related behaviors were observed in the Neuron ALKBH5 cKO and EC ALKBH5 cKO mice, apart from the evidence that Neuron cKO mice produced anxiety-like behaviors in EPM and LD (Fig. 2), which is consistent with a previous report[42]. Based on our investigations into the respective effect of astrocyte and neuron m6A methylation, it is prospective to further our research into the coordination effect of astrocyte and neuron m6A methylation on different brain regions in depression.

Glutamate contributes to the pathogenesis of depression when it is imbalanced[43]. It is elevated to a considerable level in the plasma and PFC of MDD patients[44,45]. When excessive around cells, in mechanism, it will over-activate neuronal glutamate receptors, leading to neuronal atrophy and excitotoxic neuronal death[46]. Normally, approximately 95% of the total glutamate in the forebrain can be cleared by GLT-1[47]. Therefore, deletion of GLT-1 in mice decreases glutamate uptake and consequently induces depressive-like behaviors[48]. In our study, we found astrocytic GLT-1 was subjected to m6A modification by ALKBH5, which interfered with GLT-1 translation and protein expression and then affecting the function of glutamate transport (Figs. 4 and 5). Similar to a previous report and our study[44,49], the glutamate levels in the control animals in the FIT in our study were substantially increased after the CSDS and under stress. In our further experiments, moreover, ALKBH5 cKO was also observed to have enhanced GLT-1 expression and improved glutamate uptake (Figs. 5 and 6).

It is reported excessive glutamate release is triggered by maladaptive stress responses[50], as such responses increase the activation of glutamate receptors, thereby resulting in excitotoxicity and impairing synaptic transmission[51,52]. Some studies and our patch-clamp data showed that the frequency of sEPSCs and mEPSCs in the mPFC pyramidal neurons was decreased under stress (Fig. 7)[29,53]. Another study demonstrates that the frequency of sEPSCs and mEPSCs is also reduced by knockdown of GLT-1 in the mPFC[48]. Here, we found astrocytic ALKBH5 reduction increased the expression of GLT-1 and prevented disruption of glutamatergic synaptic transmission under chronic stress (Figs. 5 and 6). Some studies show that upregulated astrocytic GLT-1 likely leads to an increase in glutamate transporter currents and efficient glutamate clearance[54]. Notably, our data

demonstrated a slight decrease in the frequency and amplitude of mEPSCs in the mPFC pyramidal neurons of Astrocyte cKO mice under basal conditions (Fig. 6). This possible discrepancy may come out of the mechanisms underlying the reduced mEPSC frequency in Astrocyte cKO and Ctrl mice with or without chronic stress. For this reason, it is worth further exploration into the effect of the coordination of these two mechanisms under stress.

SAMe constitutes the main biological methyl donor molecule involved in the RNA methylation, DNA methylation, and synthesis of neurotransmitters or phosphatidylcholine[55]. Deficiency of SAMe has been linked to psychiatric diseases such as depression[56], but it is unclear whether this association is causal. Previous clinical studies have investigated the antidepressant role of the SAMe supplementation alone or in comparison to a classical antidepressant such as imipramine[56,57]. In some animal studies, SAMe exhibits its capability of inducing antidepressant-like behaviors in FST through the catecholamine pathway and DNA methylation pathway[58,59], and it even regulates the metabolism of folic acid and methionine in the biosynthesis of neurotransmitters[60]. In other studies, SAMe is also considered as a donor of RNA methylation, participating in anxiety[42,61]. In our experiments, SAMe was reduced to a lower level in the MDD patients' plasma and depression animal models (Fig. 8), it increased m6A RNA methylation levels, produced antidepressant-like behaviors in FST and CSDS (Fig. 8) and, notably, SAMe reversed the depressive-like behaviors of astrocytic OE-Alkbh5 mice. These data indicated SAMe induces antidepressant-like effects through m6A methylation.

In summary, brain epigenomics has cellular specificity and sensitivity response to environmental stress. Astrocyte epigenetic changes critically mediate the relation between the impact of the environment and the onset and development of depressive symptoms. Astrocytic ALKBH5 functions as a regulator of depressive-like behaviors and plays an important role in response to stress.

## Methods
### Human subjects
Thirty patients with MDD (19 women and 11 men) aged 24.4 ± 6.61 (SD) years and healthy controls (20 women and 18 men) aged 25.08 ± 2.53 (SD) years were recruited from the Department of Psychology and Behavior, Guangdong 999 Brain Hospital, Guangzhou, China. None of these patients with MDD or healthy controls had taken any psychotropic medications within 4 weeks. All of the patients were clinically diagnosed by at least two psychiatrists in accordance with the Diagnostic and Statistical Manual of Mental Disorders Fourth Edition (DSM-IV) criteria and the 17-item Hamilton Rating Scale for Depression (HAMD-17). Their clinical information is detailed in Supplementary Data 1. The study with the use of peripheral blood from MDD patients was approved by the Ethics Committee of Guangdong 999 Brain Hospital, Guangzhou (NO. 2021-01-087). The participants have signed the informed consent form. There were no sex- or gender-based analysis have been performed in our study. The research was performed in accordance with the Declaration of Helsinki (ethical principles for medical research involving human subjects).

### Animals
Four to five C57BL/6J mice or CD1 mice were housed in an EVC cage (300 × 170 × 120 mm) at 23 ± 1 °C, humidity 40% under standard laboratory conditions with a 12 h light/dark cycle (lights on from 8:00 a.m. to 8:00 p.m.) and with free access to food and water. C57BL/6J mice (aged 8–12 weeks) were obtained from the Southern Medical University Animal Center (Guangzhou, China). The adult male CD1 mice (CD-1®(ICR)Mice, NO. VM0011) (older than 5 months of age) were purchased from Charles River Laboratories (Beijing). Before the behavioral tests, the C57BL/6 J mice were handled every day for 3 days, and double-blind behavioral experiments were performed between 1:00 p.m. and 5:00 p.m. All of the experiments were conducted in

accordance with the Regulations for the Administration of Affairs Concerning Experimental Animals (China) and were approved by the Southern Medical University Animal Ethics Committee (NO. 2016104).

*Alkbh5^{loxP/loxP}* mice were purchased from the Shanghai Model Organisms Center (No. NM-CKO-190004). *CaMKIIα-creER^{T2}* mice were purchased from Jackson Laboratories (Stock No. 012362). *Fgfr3-iCreER^{T2}* mice (C57BL/6 J background) were generously provided by William D Richardson (University College London). *Cdh5-CreER^{T2}* mice were purchased from Biocytegen (Stock No. 110140). *Fgfr3-iCreER^{T2}; Alkbh5^{loxP/loxP}* mice were generated by crossing the *Alkbh5^{loxP/loxP}* mice with the *Fgfr3-iCreER^{T2}* mice and so it was with the *CaMKIIα-creER^{T2}; Alkbh5^{loxP/loxP}* mice by crossing the *Alkbh5^{loxP/loxP}* mice with the *CaMKIIα-creER^{T2}* mice. *Cdh5-CreER^{T2}; Alkbh5^{loxP/loxP}* mice by crossing the *Alkbh5^{loxP/loxP}* mice with the *Cdh5-CreER^{T2}* mice. To excise the *loxP* sites by *Cre* recombination, 2-month-old male mice were i.p. injected with TAM (Sigma-Aldrich: T5648) with $100\,mg\,kg^{-1}$ of body weight once a day for 5 consecutive days. TAM was dissolved in corn oil (Sigma-Aldrich: C8267) at a final concentration of $10\,mg\,mL^{-1}$. The littermate *Alkbh5^{loxP/loxP}* mice injected with TAM were used as the controls. Only male mice (8–13 weeks old) with normal appearance and weight were used for all tests. Behavioral tests were conducted 28 days after the first TAM injection by experimenters who were blinded to the experimental group.

### Isolation of brain astrocytes and flow cytometry
Mice were anesthetized with pentobarbital sodium and perfused with ice-cold sterile PBS, followed by dissecting and coarsely chopping their brains. Then, the cells from the chopped brains were dissociated for 30 min at 37 °C in 5 mL EBSS solution containing $2\,mg\,mL^{-1}$ Papain (Sigma), $1\,mg\,mL^{-1}$ L-cysteine (Sigma), $0.5\,mg\,mL^{-1}$ EDTA (Sigma), and $100\,\mu g\,mL^{-1}$ DNase I (Sigma) for incubation. In the middle of the incubation, the brain tissues were dissociated with an 800 μm pipette, with the cell pellet resuspended in 5 mL of a 26% Percoll solution and centrifuged at $900 \times g$ (4 °C) for 15 min to enrich the astrocytes. After the cell pellet was passed through a 40-μm mesh, the cells were incubated in FACS buffer with antibody for 30 min on the ice at 4 °C in the dark and FACS analysis. The cell surface markers were stained for 30 min at 4 °C with FcR blocking reagent (Miltenyi) and treated with ACSA-2-PE of the FACS antibodies (Miltenyi Biotec, 130-102-365). Data collection was performed on a BD LSRFortessa X-20, and subsequent analysis was completed using FlowJo software.

### Simple Western
Mice were euthanized after CSDS, and the astrocytes were isolated. The astrocytes were lysed in RIPA with 1% protease inhibitor (PMSF). The samples were then mixed with Simple Western sample buffer and standards to a final concentration of $1\,\mu g\,\mu L^{-1}$. For protein detection, we used the Simple Western system (Protein Simple, Bio-Techne). Target proteins were identified with primary antibodies against ALKBH5 (Sigma: HPA007196, 1:200) and GAPDH (Proteintech: 60004, 1:500) and subsequent immunodetection using horseradish peroxidase (HRP)-conjugated secondary antibody and chemiluminescent substrate.

### Magnetic-activated cell sorting (MACS)
We isolated microglial and neuronal cellular populations utilizing MACS column-based protocols according to the manufacturer's instructions (Miltenyi Biotec: 130-107-677). The cortical tissue was enzymatically and mechanically lysed at 37 °C for 35 min to obtain a cell suspension. Then, the suspension was passed through a 40-μm cell strainer to remove debris and ensure a single-cell suspension. After a 10 min centrifugation at $300 \times g$, the pellet was gently resuspended with ice-cold PBS with 0.5% BSA buffer and incubated for 15 min at 4 °C with Myelin Removal Beads II (Miltenyi Biotec). After washing with PBS +

0.5% BSA and 10 min centrifugation at $300 \times g$, the pellet was resuspended with PBS + 0.5% BSA. The single-cell suspension was applied to a column placed on a magnetic stand (LS column and magnets from Miltenyi Biotec) to deplete the myelin fragments by magnetic separation and washed with PBS + 0.5% BSA. Finally, the cells were centrifuged for 5 min at $300 \times g$ and the pellet was resuspended with 1 mL of PBS with 0.5% BSA. After obtaining a myelin-free single-cell suspension, the microglia were labeled with anti-CD11b immunomagnetic beads (Miltenyi Biotec: 130-093-634) and isolated using magnetic columns. To isolate the oligodendrocytes, the cell suspension depleted of microglia cells was incubated with anti-oligodendrocytes immunomagnetic beads (Miltenyi Biotec: 130-094-543).

### Virus injection
The 2-month-old mice were anesthetized with sodium pentobarbital ($50\,mg\,kg^{-1}$, i.p. injection) for unilateral and fixed in a stereotactic frame (RWD) before surgery. Then they were treated with a bilateral stereotaxic injection of virus into the mPFC (AP: +1.75; ML: ±0.3; DV: −2.7 mm relative to bregma; AP, ML, and DV denote anteroposterior, mediolateral and dorsoventral distances from bregma, respectively). The coordinates were measured from bregma according to the mouse atlas. The virus (300 nL) was injected into each location at a rate of $100\,nL\,min^{-1}$. After each injection, the needle was left in place for 6 min and then slowly withdrawn.

The virus purchased from Taitool Bioscience containing iCre under a gfaABC1D promoter (AAV2/9-gfaABC1D-eGFP-iCre-SV40pA) or control virus (AAV2/9-gfaABC1D-eGFP-bGHpA) was bilaterally injected into the mPFC of *Alkbh5^{loxP/loxP}* mice to specially knockdown ALKBH5 in the astrocytes.

Then, to overexpress ALKBH5 in the astrocytes, the virus (AAV2/8-nEf1α-DIO-ALKBH5-EGFP-WPRE-PA) or control virus (AAV2/8-nEf1α-DIO-EGFP-WPRE-PA) was bilaterally injected into the mPFC of *Fgfr3-iCreER^{T2}* mice, followed by administering TAM at 5 days after virus injection.

The viruses AAV2/9-CaMKIIα-hM4D(Gi)-mCherry-WPRE-pA, AAV5-gfaABC1D-mCherry-5'miR-30a-shRNA(GLT-1)-3'miR-30a-WPREs, and AAV2/9-hSyn-eGFP-iCre-WPRE-pA were purchased from BrainVTA (Wuhan, China). The shRNA sequences targeting GLT-1 were GCTCTCACTGACTGTGTTT.

For sparse labeling of pyramidal neurons in the mPFC (AP: −1.75; ML: ±0.3; DV: −2.5 mm) of *Fgfr3-iCreER^{T2}; Alkbh5^{loxP/loxP}* cKO and control mice, the mice were bilaterally injected with a total 400 nL viral cocktail (1:1) of AAV2/9-CaMKIIα-FLP-WPRE-PA and AAV2/9-nEf1α-FDIO-EYFP-WPRE-PA. Two weeks after the injection, both Astrocyte cKO and Ctrl mice were subjected to 10 days of CSDS or non-CSDS. The brain sections were collected for confocal imaging. The basal dendritic complexity from the cell body to the surrounding 120 μm and the spine density of pyramidal neurons were analyzed with Imaris 8.0 by double-blind investigators.

### A designer receptor exclusively activated by a designer drug (DREADD)
*Fgfr3-iCreER^{T2}; Alkbh5^{loxP/loxP}* mice were bilaterally injected with hM4Di into the mPFC following TAM injection one week later. After 4 weeks of virus expression, mice were injected with clozapine-*N*-oxide (CNO; $2\,mg\,kg^{-1}$, i.p.; Sigma-Aldrich) and placed into the test room to habituate for 60 min. Stimulation-induced behaviors were scored manually by a human observer inspecting the videos post hoc (EthoVision by Noldus) while blinded to the underlying conditions.

### Fiber-photometry
Following injection of the virus (300 nL) (AAV2/9-iGluSnFR(A184S)-WPRE-hGHpA) or (AAV2/9-hSyn-GCaMp6s-WPRE-hGHpA) into the mPFC, a fiberoptic implant was advanced and secured at the same location of *Fgfr3-iCreER^{T2}; Alkbh5^{loxP/loxP}* mice.

To record fluorescence signals, a laser beam from a laser tube (488 nm) was reflected by a dichroic mirror focused by a 10× (NA of 0.3) lens and coupled to an optical commutator. Continuous video and fiber photometry acquisition were recorded during the FIT with an aggressive CD1 mouse. The fluorescence was bandpass filtered (MF525-39, Thorlabs) and collected by a photomultiplier tube (R3896, Hamamatsu). An amplifier (C7319, Hamamatsu) was used to convert the photomultiplier tube current output to voltage signals, which were further filtered through a low-pass filter (40 Hz cutoff; Brownlee 440). To minimize photobleaching, the laser power at the fiber tip was adjusted to 30 μW.

Bulk fluorescence signals were acquired and analyzed with MATLAB software. The z-score for a population of astrocytes was calculated using the following formula: $\text{z-score} = (F_{Signal} - F_{Basal})/\text{STD}(F_{Basal})$.

## Cell culture
Tissues were isolated from the cerebral cortex of postnatal day 1 mouse washed in ice-cold PBS and transferred to a 50 mL Falcon tube containing 0.5 mL of PBS for dissociation using a pair of sterile operating scissors. Then, they were incubated with 0.25% trypsin (Gibco) in 0.5 mM EDTA at 37 °C for 10 min. In the following step, culture medium was added to inhibit the reaction, and the cell suspension was transferred into 15 mL tubes and centrifuged at $900 \times g$ for 6 min. The pellet was resuspended in 10 mL of culture medium (for neurons: Neurobasal Medium-A + 1% B27 + 1% Glutamax; for astrocytes, DMEM F12 + 10% FBS). Finally, the cells were placed in a culture flask at a density of $5 \times 10^6$ cells per 5 mL and incubated in a humidified incubator containing 5% $CO_2$ air at 37 °C.

Cultures of mPFC glia and neurons were prepared from E18 mice following previously described methods. Briefly, cells were plated at a density of $60 \times 10^3$ cells per mL on poly-L-lysine precoated coverslips. Cultures were kept in a neurobasal medium (Invitrogen) with 3% horse serum (Invitrogen) for several days before changing to serum-free neurobasal medium (Invitrogen). Cultures were maintained at 37 °C in 5% $CO_2$ for 20 d in vitro at maximum.

## Western blotting analysis
Cultured cells and mouse brain tissues were homogenized in lysis buffer (RIPA) in the presence of protease inhibitor (PMSF) on ice for 30 min and centrifuged at $10,000 \times g$ for 20 min at 4 °C. Protein samples (30–60 μg) were separated using 10% SDS-PAGE gels and subsequently immunoblotted onto polyvinylidene difluoride (PVDF) membranes (Millipore) in ice-cold buffer (25 mM Tris HCl, 192 mM glycine, and 20% methanol) by electrotransfer for 2 h. The membranes were blocked with 5% nonfat milk powder dissolved in TBST and then incubated with the indicated antibodies at 4 °C overnight (ALKBH5, Sigma: HPA007196, 1:1000; GLT-1 Abcam: Ab205247, 1:1000; GAPDH, Proteintech: 60004, 1:5000). Horseradish peroxidase (HRP)-conjugated goat-anti-rabbit (Abclonal: AS014, 1:5000) and goat-anti-mouse Abclonal: AS003, 1:5000) secondary antibodies were incubated at room temperature for 1 h. Blots were detected using enhanced chemiluminescence (Pierce). Protein abundance was quantified by analyzing the Western blot bands using Image Lab (Bio-Rad) software. Quantified band intensities were normalized to GAPDH levels and averaged from at least three independent experiments.

## ELISA
The mice were anesthetized with sodium pentobarbital. The transverse mPFC was cut from each mouse brain. The tissues were washed in ice-cold PBS and then homogenized in fresh lysis buffer. The homogenates were centrifuged for 5 min at $10,000 \times g$. The supernates were immediately collected and assayed according to the manufacturer's protocols. The SAMe ELISA kits were purchased from Dogesce (Human: DG94114Q; Mouse: DG91579Q).

For the m6A RNA methylation analysis, the SAMe ELISA kits were purchased from Epigentek (P-9005-96).

## Immunofluorescence staining
Animals were anesthetized and perfused with saline followed by 4% PFA in 0.1 M PBS, pH 7.4. The brains of the animals were removed, post-fixed overnight in 4% PFA at 4 °C and transferred to 30% sucrose in 0.1 M PBS, pH 7.4, followed by cutting the coronal sections (40 μm) on a freezing microtome (Leica CM3050 S). After being seeded on coverslips to the appropriate density, the cultured cells were fixed in 4% paraformaldehyde for 15 min and permeabilized in 0.5% Triton X-100 solution for 5 min at room temperature. Then, the cells and sections were washed with PBS three times and incubated first in a blocking buffer containing 3% bovine serum albumin in 0.2% Triton X-100/PBS for 2 h at room temperature and then with primary antibodies against ALKBH5 (1:300; Sigma: HPA007196), NeuN (1:300; Millipore: MAB377), Iba1 (1:500; Abcam: ab178846), Sox10 (1:300; Abcam: ab155279), and S100β (1:300; Abcam: ab52642) in blocking buffer overnight at 4 °C. After washing the treated cells and sections with PBS again for three times, they were further incubated with Alexa Fluor 488- or Alexa Fluor 568-conjugated secondary antibodies at room temperature for 2 h. Alexa fluor 488-anti-Mouse secondary antibody (Invitrogen, A32723, 1:500), Alexa fluor 568-anti-Mouse secondary antibody (Invitrogen, A11004, 1:500), Alexa fluor 488-anti-Rabbit secondary antibody (Invitrogen, A32731, 1:500), Alexa fluor 568-anti-Rabbit secondary antibody(Invitrogen, A11036, 1:500). The nuclei were counterstained with DAPI and the coverslips were mounted onto glass slides with anti-fade solution and visualized using a Nikon fluorescence microscope (Nikon Instruments Inc.).

## Real-time quantitative PCR
The total RNA from the whole blood and brain tissues was extracted using TRIzol (Thermo Fisher Scientific) in line with the manufacturer's instructions, and the RNA samples were stored at −80 °C. Quantitative PCR was performed by using the Transcriptor First Strand cDNA Synthesis kit (TAKARA) and TB Green® Premix Ex Taq™ II (TAKARA) on a LightCycler 96 Real-Time System (Roche). Differences in gene expression were calculated by the $2^{-\Delta\Delta CT}$ method and were presented as the fold change. The relevant primers was provided in the Supplementary Data 5.

## Plasmid construction and cell transfection
dCas9-m Cherry and GLT-1-targeting-sgRNA-puro-eGFP plasmids (BrainVTA) were constructed and transfected into primary cultured astrocytes to alter the 3' UTR region 201 m6A modification site in GLT-1. Then, the astrocytes were transfected using TransIT-X2 (Mirus Bio) for 24 h at 37 °C according to the manufacturer's protocol.

## Dual-luciferase assays
The SLC1A2-3' UTR WT was PCR-amplified directly from mouse cDNA and then cloned into the 3' end of the Renilla luciferase gene of the psiCHECK-2 dual-luciferase vector containing firefly luciferase driven by a different promoter as an internal control (Promega, Madison, WI, USA) via the Xho I and Not I sites. Mutant reporter constructs (SLC1A2-3' UTR-A, SLC1A2-3' UTR-C) were obtained from the SLC1A2-3' UTR WT reporter using a QuikChange site-directed mutagenesis kit (Stratagene, La Jolla, CA). Primary astrocytes were cultured, and transfection using Lipofectamine 3000 reagents (Life Technologies), and dual-luciferase activity (Promega) was measured with the Wallac Victor V 1420 Multilabel Counter (PerkinElmer, San Jose, CA, USA) to yield the ratio of Renilla luciferase activity to firefly luciferase activity. All luciferase readings were taken from more than three individual biological repeats.

## Evaluation of glutamate uptake ability
L-Glutamate (200 μM) (Sigma) was added, and glutamate uptake ability was determined by measuring glutamate levels in the medium after

4 h. The astrocytes treated with rLV-GFAP or Ctrl virus in each group were treated with 200 μM L-glutamate and incubated for 4 h at 37 °C. At each time point, the concentration of glutamate in the medium was measured with a glutamate assay kit (Abcam: ab83389) according to the manufacturer's protocol, and the glutamate uptake ability was calculated by (Concentration $_{4h}$ - Concentration $_{0h}$)/ Concentration $_{0h}$ × 100%.

## Electrophysiological recording

The mice were anesthetized with pentobarbital sodium. Their brains were quickly removed and cut into 300-μm horizontal slices containing the mPFC using a Leica LS1200s vibrating microtome. which were prepared in ice-cold oxygenated cutting artificial cerebrospinal fluid (aCSF) containing (in mM) 220 Sucrose, 2.5 KCl, 1.3 $CaCl_2$, 2.5 $MgSO_4$, 1 $NaH_2PO_4$, 26 $NaHCO_3$, and 10 Glucose. After that, the slices were transferred to a chamber with oxygenated recording aCSF containing (in mM) 126 NaCl, 26 $NaHCO_3$, 3.0 KCl, 1.2 $NaH_2PO_4$, 2.0 $CaCl_2$, 1.0 $MgSO_4$, and 10 Glucose and held at 32 °C for 30 min and then at room temperature for 1 h. The mPFC pyramidal neurons were viewed with a 40× water immersion objective (Zeiss). Recording electrodes with a resistance of 3–5 MΩ were pulled from borosilicate glass capillaries.

To record spontaneous excitatory postsynaptic current (sEPSC), the internal pipette solution for the recording electrodes contained (in mM) 125 cesium methanesulfonate, 5 CsCl, 10 HEPES, 0.2 EGTA, 1 $MgCl_2$, 4 Mg-ATP, 0.3 Na-GTP, 10 phosphocreatine and 5 QX314 (pH 7.40, 290 mOsm). Recordings were made with a HEKA EPC10 amplifier with the signals filtered at 5 kHz and digitized at 10 kHz. Finally, the sEPSCs with 20 μM bicuculine (Sigma) added to the cerebrospinal fluid in the voltage clamp (Vclamp = −70 mV) and the mEPSCs in the presence of 1 μM tetrodotoxin (Aladdin) and 20 μM bicuculine (Sigma) were recorded.

The action potentials were assessed under the current-clamp mode, with the pipette solution including (in mM): 125 K-gluconate, 5 KCl, 10 HEPES, 0.2 EGTA, 1 $MgCl_2$, 4 Mg-ATP, 0.3 Na-GTP, and 10 phosphocreatine (pH 7.40, 285 mOsm). A series of depolarizing currents (from 0 to 200 pA, at a step of 10 pA) were injected to induce action potentials.

## Behavioral studies

**Chronic social defeat stress.** For the CSDS protocol, the adult male CD1 mice (older than 5 months of age) were purchased from Charles River Laboratories (Beijing). An experimental C57BL/6J intruder mouse was exposed to a CD1 aggressor mouse for 10 min. After the 10-min defeat session, the CD1 mouse and the intruder C57BL/6J mouse were kept in the same cage but separated by a divider for the remainder of the day. This procedure was repeated for 10 consecutive days, using a different aggressor CD1 mouse every day.

**Subthreshold social defeat stress.** This procedure was identical to the normal chronic social defeat stress procedure, with the exception that the procedure lasted for three consecutive days.

**Female chronic social defeat stress.** Following previous studies[62], we treated experimental female mice with the urine of a male CD1 mouse unknown to the resident CD1 aggressors, then exposed the female mice to a different resident aggressive mouse for 5 to 10 min each day for 10 days.

**Social interaction test.** The long-term behavioral consequences of the CSDS were tested by SI 24 h after the CSDS. Social avoidance behavior was measured according to a two-stage social interaction test. In all behavioral experiments, the animals' tracks were monitored with a video tracking system. The mice were placed in a novel cage ($44 × 44 × 44$ cm³) containing an empty metal cage ($9.5 × 9.5 × 8$ cm³), and their movements were tracked for 2.5 min in the absence of the aggressor. Their movements were then followed for 2.5 min in the presence of the caged aggressor. The apparatus was cleaned with a solution of 70% ethanol in water to remove olfactory cues following each trial, and all of the behavioral tests were conducted in the dark. A SI index was calculated (time spent in the interaction zone in the presence versus the absence of a target mouse). The mice were considered susceptible when their scores were <1, and resilient when their scores were ≥1.

**Forced interaction test.** The test mice were placed in a protective wire-mesh enclosure ($13 × 7 × 9$ cm) in the home cage of the aggressor. One of the mice was placed in an 8 cm × 15 cm Plexiglas cylinder. Following a 5-min recording period during which fluorescence signals were recorded, a CD1 aggressor mouse was introduced into the cage outside of the cylinder (18 cm high walls). Fluorescence signals were then recorded for an additional 5 min.

**Forced swimming test.** The FST was performed in a glass cylinder (height 45 cm, diameter 19 cm), which was filled to 23 cm with water (22–24 °C). The mice were placed in the cylinder. The test lasted for 6 min. The mice were adapted in the first 2 min, and the duration of immobility was recorded during the final 4 min. The immobility of the mice was recorded by Ethovision XT (Noldus, USA) software.

**Open field test.** The mice were placed in an open chamber ($40 × 40 × 30$ cm) that was made of gray polyvinyl chloride. The mice could move freely in it. The mice were gently placed in the center and explored the area for 5 min. The total distance traveled and time spent in the center ($20 × 20$ cm²) were recorded by a VersaMax Animal Activity Monitor system and analyzed by VersaMax 4.20 software.

**Light-dark box test.** In the LD, a black box ($40 × 20 × 12$ cm³) was placed on one side of the open field test chamber ($40 × 40 × 30$ cm³), which divided the chamber into two compartments of equal size. The mice were introduced into the dark chamber and allowed free exploration for 5 min. The duration in each area and the first latency and total entries to the dark compartment were detected by a VersaMax Animal Activity Monitor system and analyzed by VersaMax 4.20 software.

**Elevated plus maze.** The apparatus consisted of two opposing open arms ($30 × 5 × 0.5$ cm³) and two opposing enclosed arms ($30 × 5 × 15$ cm³), which were connected by a central platform ($5 × 5$ cm²) and positioned 50 cm above the ground. The behavior was tracked for 5 min with an overhead camera and EthoVision 11.0 software (Noldus).

**Rotarod test.** The mice were placed on a stationary rotarod (AccuRotor Rota Rod Tall Unit, 63 cm fall height, 30 mm diameter rotating dowel). The dowel was accelerated to 60 RPM in 5 min, and the latency to fall (in seconds) was recorded.

**Differential gene expression analysis.** We analyzed the external, publicly available Gene Fpkm data from 248 samples, which were downloaded from GEO (GSE102556) and were preprocessed to transform Fpkm to Tpm format in the limma package. The limma Bioconductor package was then used to analyze the gene expression data. Differentially expressed genes (DEGs) between the groups were determined in several stages to ensure both statistical significance and biological relevance. DEGs were assessed through a generalized linear model implemented in limma, with the phenotype (MDD versus CTRL) as main factors for BA8_9 brain region. An individual gene was called differentially expressed if the P value of its t-statistic was at most 0.05.

**Library preparation and high-throughput sequencing.** Total RNA was isolated from the tissues using the Magzol Reagent (Magen, China)

according to the manufacturer's protocol. The quantity and the integrity of RNA yield were assessed by using the K5500 (Beijing Kaiao, China) and the Agilent 2200 TapeStation (Agilent Technologies, USA), respectively. m6A antibody immunoprecipitation RNA was quality control with Qubit (Thermo Fisher Scientific, USA) and Agilent 2200 TapeStation (Agilent Technologies, USA). Briefly, the RNA were fragmented to approximately 200 bp. Then, the RNA fragments were subjected to first strand and second strand cDNA synthesis following by adapter ligation and enrichment with a low-cycle according to instructions of NEBNext® Ultra RNA LibraryPrep Kit for Illumina (NEB, USA). The final library product was assessed with Agilent 2200 TapeStation and Qubit® (Life Technologies, USA) and then sequenced on Illumina (Illumina, USA) platform with pair-end 150 bp at Ribobio Co. Ltd (Ribobio, China).

**Data preprocessing.** Adapter and low-quality bases were trimmed with Trimmomatictools(version:0.36), and the clean reads underwent rRNA deleting through RNAcentral to get effective reads. Genomic alignment (version from UCSC genome browser) was using Tophat(version:2.0.13) to get uniquely mapping reads.

**Expression calculation.** Effective reads from the input samples was used for RNA-seq analysis, and the reads count value of each transcript was calculated by HTSeq(version:0.6.0). Differential expression genes were identified by the DESeq2 R package according to the criteria of $|\log_2(\text{Fold Change})| \geq 1$ and $q$ value $< 0.05$. The relevant supplementary tables was uploaded in the Supplementary Data 2.

**Peaks calling and motif identification.** MACS2 (version 2.1.0.20151222) was employed to perform m6A peak calling. Then Homer (version:4.8) was used to annotate the m6A Peaks. The nucleotides in the m6A Peaks region were used for detection of the consensus m6A motif by DREME (version:4.11.1) and MEME(version:4.11.1). Motif central enrichment was performed by CentriMo (version:4.11.1). Differential methylation was determined by MetDiff according to the criteria of $|\log_2(\text{Fold Change})| \geq 2$ and $q$ value $< 0.05$. The relevant supplementary tables was uploaded in the Supplementary Data 3.

**Functional enrichment analysis.** Gene ontology (GO) and Kyoto Encyclopedia of Genes and Genomes (KEGG) pathway enrichment analysis were performed using ClusterProfiler R package/ KOBAS3.0. The results from the enrichment analysis were restricted to GO biological process and KEGG pathway terms, with an adjusted $P$ value $< 0.05$ considered to be significant.

**Quantification and statistical analysis.** All experiments and data analyses were conducted blindly. Statistical comparisons were performed using SPSS 20.0 software with appropriate inferential methods, with t data presented as the mean ± s.e.m. The normally distributed data were tested by a two-sided unpaired $t$-test for two-group comparisons and one- and two-way analysis of variance (ANOVA) followed by Bonferroni's test for multiple comparisons. The non-normally distributed data were analyzed by Mann–Whitney $U$-tests for two-group comparisons and by the Kruskal–Wallis test and Dunn's multiple comparisons test for more than two groups. Statistical significance was set at $^*p < 0.05$, $^{**}p < 0.01$, $^{***}p < 0.001$, and $^{****}p < 0.0001$. n.s. no significance.

**Study approval.** All of the animal experiments were conducted in accordance with the Chinese Council on Animal Care Guidelines, and ethics approval was obtained from the Research Ethics Board at Southern Medical University. The use of MDD patients' peripheral blood was approved by the Ethics Committee of Guangdong 999 Brain Hospital, Guangzhou (NO. 2021-01-087). The research was performed in accordance with the Declaration of Helsinki (ethical principles for medical research involving human subjects). Informed consent was obtained from all individual participants included in the study. Authors are responsible for the correctness of the statements provided in the manuscript.

## Reporting summary

Further information on research design is available in the Nature Portfolio Reporting Summary linked to this article.

## Data availability

The MeRIP-seq data generated in this study have been deposited in the Genome Sequence Archive under accession code CRA012006 (https://ngdc.cncb.ac.cn/gsa)[63,64]. Previously described data was downloaded from the GEO database (GSE102556). Source data are provided with this paper.

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

## Acknowledgements

We thank William D. Richardson (University College London, London, UK) for providing the *Fgfr3-iCreER^{T2}* mice. We thank Yi-Da Pan. and

L.Huang for helping with the patch-clamp experiments. We thank Zhi-ying Deng for helping with the dual-luciferase assays. We thank Sheng Zhang and Rui-yuan Pan for helping with the MACS experiments. We thank Rang-Ke Wu for helping edit the manuscript. This work was supported by the National Natural Science Foundation of China (Grant No. 32271062 to X.C.), STI2030-Major Projects 2022ZD0204700, Guangzhou Key Research Program on Brain Science (202206060001 to X.C.), Guangzhou Science and Technology Project (202007030013 to T.-M.Gao); the Key-Area Research and Development Program of Guangdong Province (2018B030334001); Guangdong-Hong Kong Joint Laboratory for Psychiatric Disorders (2023B1212120004).

## Author contributions

X.C. and F.G. designed the study and wrote the paper. X.C., F.G., and J.-M.L. analyzed the data. F.G. performed most of the experiments. J.-M.L., J.F., F.G., and P.-L.K. performed behavioral experiments with the help of Y.-Y.F. as well as stereotactic injection and in vivo recordings with the help of C.-L.L. and J.R. J.F., F.G., and L.-Y.C. performed the western blotting. Q.-L.Z. and J.-W.M. performed the FACS and Simple Western analysis. F.G. performed the MACS. J.R. and S.-J.L. were responsible for cell culture. F.G. and C.Z. performed the patch-clamp experiments. Y.-L.W. and T.-T.G. collected the blood of MDD patients and healthy controls. F.G. and J.-M.L. performed the qPCR analysis. J.-M.L. and H.-T.J. carried out genotyping. T.-M.G. and B.-X.P. reviewed and edited the manuscript. X.C. supervised the project all through the phases.

## Competing interests

The authors declare no competing interests.
