## [Peer Review File · Nature Communications]

Astrocytic ALKBH5 in stress response contributes to depressive-like behaviors in mice.REVIEWER COMMENTS

Reviewer #1 (Remarks to the Author):

Guo et al. investigated how m6A modification was associated with behavioral alterations related to major depressive disorders (MDD), with a particular focus on astrocytes. The authors first discovered an upregulation of ALKBH5, an RNA demethylase, in MDD patients as well as in depression model mice. Using several CreERT2 mouse lines, the authors deleted *Alkbh5* from different cell types and found that *Alkbh5* deletion in astrocytes presented anti-depressant effects under chronic social defeat stress (CSDS). Consistently, overexpression of *Alkbh5* in astrocytes worsened stress-induced behavioral changes. The authors further showed that astrocytic *Alkbh5* deletion increased the m6A methylation of GLT-1, an abundant astrocytic glutamate transporter, and therefore was beneficial to maintain synaptic transmission, neuronal morphology, and neuronal activities. Finally, the authors showed that treatment with SAME ameliorated stress-induced behavioral phenotypes.

Both m6A modification and ALKBH5 have been previously documented to be associated with MDD. Consistently, a recent study by Huang et al. (*Biological Psychiatry*, 2020) identified circSTAG1/ALKBH5/m6A methylation pathway as a key regulator of astrocyte function and depressive-like behavior in a chronic unpredictable stress-induced depression model. In this current study, the authors provided additional evidence to functionally link astrocytic m6A methylation to neuronal function and depressive-like behaviors, which were largely unclear. Thus, the work by Guo et al. is of importance from this perspective. However, some of the conclusions drawn from the data are open to interpretation and there are several major issues that the authors need to adequately address. Without further supporting evidence, the current conclusions are unconvincing.

Major points:

1. Expression of *Alkbh5* is enriched not only in astrocytes but also in oligodendrocytes, endothelial cells, and fibroblast cells etc. Therefore, the authors should include experiments to validate cell-type specificity of *Alkbh5* deletion as well as *Alkbh5* overexpression are necessary to appropriately interpret their effects on depressive-like behaviors. These experiments are particularly important given the fact that *Fgfr3-iCreERT2* mouse line was found to drive recombination in all major interneuron subtypes in the olfactory bulb (Young et al. *Glia*, 2010). If this is the case, then *Alkbh5* deletion in olfactory interneurons may likely disrupt CSDS-induced behavioral changes. Similarly, astrocyte-specific deletion of *Alkbh5* in the mPFC with AAV9-GfaABC1D-iCre should also be validated because AAV-mediated Cre recombination has been found to have substantial neuronal contaminations.
2. Cortical astrocytes are well-known to have extremely low levels of GFAP (Middeldorp et al. *Progress in Neurobiology*, 2011). However, immunohistochemistry images from Figures 2 and 3 showed significant GFAP staining in mPFC astrocytes even from control mice. This raised a concern of astrocyte reactivity in the mouse models being used that resulted in altered behavioral phenotypes. The authors should provide additional evidence to rule out this possibility.
3. It is interesting that *Alkbh5* deletion in astrocytes decreased m6A levels but increased m6A methylation of GLT-1. How did this happen? The authors suggested that observed phenotypes of synaptic transmission (Figure 6), neuronal morphology and neuronal activities (Figure 7) were caused by increased GLT-1 expression in the absence of astrocytic ALKBH5, if correct, then astrocyte-specific knockdown of GLT-1 with RNAi or CRISPR-Cas9 should reverse these effects. The authors should include additional experiments to test their hypothesis.
4. SAME was applied to enhance m6A modification as well as ameliorate depressive-like phenotypes induced by CSDS. However, direct evidence showing increased m6A methylation by SAME was missing. If overall methylation was indeed increased, does this change also apply to GLT-1?

Minor points:

1. There were misuses of scientific terms. For instance, the trisynaptic circuit refers to a specific neural circuit in the hippocampus, including neurons from three different hippocampal regions. In contrast, tripartite synapses refer to the structures involving astrocytes and neurons.
2. Typos and grammatical mistakes were present throughout the manuscript. The authors should check carefully and correct all the mistakes.
3. Providing a full list of genes identified in Figure 4 would be useful to the readers.
4. In Figure 5, the authors altered methylation modification sites of GLT-1 with the CRISPR-Cas9 system in cultured astrocytes. However, whether the downregulation of GLT-1 was directly caused by reduced methylation of GLT-1 was unclear.
5. Missing X axis label of currents in Figure 6c.

Reviewer #2 (Remarks to the Author):

In this paper, the authors delve into an exploration of cell types and mechanisms through which RNA modifications contribute to depressive-like behaviors. Their research highlights the prominence of Alkbh5, an RNA demethylase, which exhibits heightened levels in patients diagnosed with major depressive disorder (MDD) as well as in depression-induced mouse models. Notably, Alkbh5 manifests greater susceptibility to stress in astrocytes compared to neurons and endothelial cells. The paper reveals that manipulation of Alkbh5 activity in astrocytes within the medial prefrontal cortex of mice exerts a modulatory influence on depressive-like behaviors. A pivotal finding attributes the primary impact of Alkbh5 to the glutamate transporter GLT-1. The experimental methodologies employed demonstrate a commendable incorporation of cutting-edge techniques. Below, several points and comments are presented for your consideration:

1. An intriguing observation pertains to the discernible decrease in FTO mRNA in both blood samples from MDD patients and the CSDS mouse model of depression. Despite FTO mRNA elevation in the PFC of Sus and Res mice alike, the pertinence of FTO expression changes in the blood warrants further inquiry. It would be valuable to explore FTO levels in the LPS mouse model of depression, thereby enhancing the comprehensiveness of the investigation.
2. A noteworthy concern emerges regarding the Fgfr3-iCre line, which possesses the potential to target not only astrocytes but also progenitor cells inclusive of those contributing to adult neurogenesis in the hippocampus (J Neurosci 2021: 2899-2910). Given this, it is prudent to contemplate the possibility that some behavioral effects could potentially stem from the manipulation of Alkbh5 activity in adult-born neurons.
3. The profound impact of CSD stress on dendritic spine density within the mPFC stands out; however, the effects on mmEPSC frequency appear somewhat subdued, possibly due to the mode of presentation. A suggestion is to illustrate the mEPSC frequency as averages rather than cumulative histograms, potentially providing a clearer representation.
4. In Figure 6 and the corresponding text, the assertion is made that "Astrocytic Alkbh5 prevents the disruption of glutamatergic synaptic transmission from social stress." The term "disruption" may not accurately reflect the minor alterations observed in electrophysiological parameters among cKO mice. An alternative description could be more aligned with the observed effects.
5. An aspect requiring clarification pertains to the duration of SAME-induced antidepressant effects in mice. Is this effect acute in nature or does it span a longer timeframe?

Minor Comments:

1. It appears that the immunostaining for Alkbh5 is weakly present in Figure 2C. Higher magnification

pictures and the usage of alternative colors might improve this presentation. Also, given that GFAP proves suboptimal as an astrocytic marker in the cortex, have the authors explored alternative markers for astrocytes?

2. Line 130 contains an inaccuracy. Neuronal and endothelial cell populations are not present in comparable numbers, as indicated.

Reviewer #3 (Remarks to the Author):

In the manuscript, entitled "Cell-specific and sensitive epitranscriptomic m(6)A of stress response in depression," Guo and colleagues present an intriguing series of results linking alterations in RNA m6A methylation – and its associated enzymatic machinery – in prefrontal cortex to stress-induced physiological and behavioral alterations associated with depression. In profiling the 'writers' and 'erasers' of m6A in both the periphery and brain of mice exposed to chronic social stress, as well as in human MDD vs. control samples, they found that ALKBH5, an RNA demethylase of m6A, is increased in its expression (mRNA and protein) in both blood and brain of MDD patients and chronically stressed male mice (in stress-susceptible, but not stress-resilient, animals). Further explorations in vitro comparing cellular stress-induced expression of Alkbh5 in cultured neurons, astrocytes and endothelial cells revealed an astrocytic-specific pattern of upregulation, which was confirmed following social stress in mPFC of mice after sorting for astrocytes specifically. Thus, they next generated mice with conditional deletion of Alkbh5 specifically in astrocytes (vs. neurons or endothelial cells) and found that such deletion only in astrocytes promoted antidepressant-like behaviors (while counterintuitively decreasing m6A epitranscriptome-wide). Bidirectional behavioral regulation was also confirmed via astrocyte specific OE studies and astrocytic restoration of Alkbh5 in KO mice. To next explore the molecular mechanism through which upregulation of Alkbh5 in astrocytes may contribute to stress-induced behavior, the authors performed a series of in vitro and in vivo Ca²⁺ imaging and electrophysiological studies to demonstrate that upregulation of Alkbh5 in response to chronic stress contributes to: increased m6A on GLT-1 mRNA (with increased expression of GLT-1 also observed), increased stress-induced glutamate uptake into astrocytes and reduced sEPSCs and mEPSCs in cortical neurons, thus suggesting that chronic stress functions, at least in part, in mPFC to alter astrocytic glutamate uptake, thereby reducing the physiological activity of glutamatergic neurons, which would be consistent with studies in both mice and humans suggesting aberrant roles for reduced mPFC activity in driving depressive-like behaviors. Overall, this is a rigorous and exciting study that implicates a novel biological process in the regulation of depressive-like states. However, there are a number of concerns that should be addressed prior to publication.

1. While the authors confirmed that Alkbh5 is upregulated specifically in astrocytes vs. neurons or endothelial cells following chronic stress, they should also explore Alkbh5's expression in other glial cells (e.g., microglia, oligodendrocytes), which may also be responsive to stress. This could be done either by performing FACS/FANS or MACS to isolate the other major glial subtypes, followed by qPCR and/or western blotting analyses.
2. While the links between increased Alkbh5 expression in astrocytes, GLT-1 m6A (and its increased expression) and disruptions in glutamate signaling in mPFC following chronic stress are very interesting, a couple more experiments need to be performed to definitively link these processes together – a) the authors should provide evidence that astrocytic OE of Alkbh5, which promotes depressive-like behaviors, results in increased GLT-1 m6A and expression, enhanced glutamate uptake into astrocytes and reduced mEPSCs in mPFC neurons (i.e., to further demonstrate the bidirectional relationship; and b) if the experiments in "a" work as expected, the authors should then pharmacologically inhibit GLT-1 under astrocytic Alkbh5 OE conditions to definitively show that the reductions in neuronal mEPSCs are caused by alterations in the function of GLT-1 specifically.
3. Similarly, the authors should attempt to demonstrate that links between increased Alkbh5 expression astrocytes and the observed alterations in neuronal mEPSCs are functionally linked to the observed behaviors. This could be done by performing defeat in WT vs. astrocytic KO mice – which

leads to reduced glutamate uptake by astrocytes and increased sEPSCs/mEPSCs in the KO animals - coupled to chemogenetic or optogenetic inhibition of glutamatergic neurons in mPFC to see if the positive effects of Alkbh5 KO in astrocytes can be reversed simply by silencing the neurons that are receiving increased glutamatergic signaling.

4. Given the prevalence of MDD in females vs. males, it would be interesting to know whether stressed females also display such increases in Alkbh5 expression in mPFC. Robust social defeat protocols now exists for females and could be employed for these assessments.

5. Are classical antidepressants, such as fluoxetine, sufficient to reverse aberrant increases in Alkbh5 expression in mPFC?

6. While the authors have chosen to focus on social interaction and forced swim tests to monitor depressive-like behaviors (as well as additional anxiety related behaviors), given roles for mPFC in driving anhedonic symptoms in depression, it will be important that additional measure of anhedonia be assessed.

7. The SAM experiments to induce m6A are highly non-specific to m6A (i.e., they will impact DNA methylation, histone methylation, etc.) or cell-type and do not fit well with the rest of the story, which is focused on cell-type and mechanism specific effects. I would suggest either moving these data to the supplement or removing them from the manuscript altogether.

8. There are some concerns over the validations of KO vs. OE effects presented, as it appears that these manipulations are not very robust in manipulating Alkbh5 in the manner in which the authors wish for them to be manipulated (e.g., the KO of Alkbh5 in astrocytes appears only very weak). Perhaps this is just a technical issue with the manner in which the authors have chosen to validate these KO/OE effects, but as it stands, additional cell-type specific validations see necessary.

9. As presented, the differential expression and MeRIP-seq data appear weak, with the results shown only employing nominal p-values of $p < 0.05$. The authors should re-run their analyses using appropriate multiple testing corrections. Additionally, while I understand the author's focus on GLT-1 for the remainder of the manuscript, however, they should attempt to strengthen these analyses and provide a broader overview of the effects being observed.

10. Finally, the data presented indicate that Alkbh5 KO decreases m6A in brain, which runs counter to Alkbh5's known role in removing m6A. Do the authors have a reasonable explanation for this?

Author response

Manuscript ID: NCOMMS-23-32659A

Title: Cell-specific and sensitive epitranscriptomic m(6)A of stress response in depression

We would like to sincerely thank you for taking the time to consider and review our manuscript and for giving us an opportunity to revise our manuscript. We have performed additional experiments to address the most of reviewer's concerns. We also have made modifications and corrections according to the comments and suggestions. The point-to-point responses to the reviewers' comments are listed in the following pages. We thank all reviewers for their constructive comments and suggestion, which have been thoroughly addressed. We hope that the ambiguities in the manuscript have now been clarified.

Reviewer 1:

Guo et al. investigated how m6A modification was associated with behavioral alterations related to major depressive disorders (MDD), with a particular focus on astrocytes. The authors first discovered an upregulation of ALKBH5, an RNA demethylase, in MDD patients as well as in depression model mice. Using several CreERT2 mouse lines, the authors deleted *Alkbh5* from different cell types and found that *Alkbh5* deletion in astrocytes presented anti-depressant effects under chronic social defeat stress (CSDS). Consistently, overexpression of *Alkbh5* in astrocytes worsened stress-induced behavioral changes. The authors further showed that astrocytic *Alkbh5* deletion increased the m6A methylation of GLT-1, an abundant astrocytic glutamate transporter, and therefore was beneficial to maintain synaptic transmission, neuronal morphology, and neuronal activities. Finally, the authors showed that treatment with SAME ameliorated stress-induced behavioral phenotypes.

Both m6A modification and ALKBH5 have been previously documented to be associated with MDD. Consistently, a recent study by Huang et al. (*Biological Psychiatry*, 2020) identified circSTAG1/ALKBH5/m6A methylation pathway as a key regulator of astrocyte function and depressive-like behavior in a chronic unpredictable stress-induced depression model. In this current study, the authors provided additional evidence to functionally link astrocytic m6A

methylation to neuronal function and depressive-like behaviors, which were largely unclear. Thus, the work by Guo et al. is of importance from this perspective. However, some of the conclusions drawn from the data are open to interpretation and there are several major issues that the authors need to adequately address. Without further supporting evidence, the current conclusions are unconvincing.

We appreciate the comments and suggestions concerning our manuscript. The comments and suggestions were very valuable and helpful for improving our manuscript. We carefully studied the comments and made corrections that we hope will be met with approval.

1. Expression of *Alkbh5* is enriched not only in astrocytes but also in oligodendrocytes, endothelial cells, and fibroblast cells etc. Therefore, the authors should include experiments to validate cell-type specificity of *Alkbh5* deletion as well as *Alkbh5* overexpression are necessary to appropriately interpret their effects on depressive-like behaviors. These experiments are particularly important given the fact that *Fgfr3-iCreER^{T2}* mouse line was found to drive recombination in all major interneuron subtypes in the olfactory bulb (Young et al. *Glia*, 2010). If this is the case, then *Alkbh5* deletion in olfactory interneurons may likely disrupt CSDS-induced behavioral changes. Similarly, astrocyte-specific deletion of *Alkbh5* in the mPFC with AAV9-gfaABC1D-iCre should also be validated because AAV-mediated Cre recombination has been found to have substantial neuronal contaminations.

Answer: Thank you for your helpful advice. As previously reported, after tamoxifen administration, ~90% of all protoplasmic and fibrous astrocytes could be labeled in the adult *Fgfr3-iCreER^{T2}* mouse brain (1, 2). *Fgfr3-iCreER^{T2}* transgenic mice are widely used in astrocyte research (3-6). However, Young et al show that *Fgfr3* is also expressed in the interneurons of olfactory bulb (1, 7). Altogether, given the limitation of these cre mice and specific changes in *Alkbh5* in the mPFC, we performed the following experiments.

First, we co-stained *Alkbh5* and S100 β in the mPFC slices of Astrocytic *Alkbh5* cKO mice. Confocal images and analysis showed that *Alkbh5* and S100 β double positive cells were significantly decreased in Astrocyte cKO mice compared to littermate control (**Figure 2**).

However, the level of *Alkbh5* expression in S100 β -negative cells remained unchanged between Astrocyte cKO and control mice. **(Extended Data Fig. 4).**

Besides, given the limitation of these cre mice and specific changes in *Alkbh5* in the mPFC astrocytes, we performed brain-region knockdown by injecting the AAV-gfaABC1D-eGFP-iCre virus, which preferentially targets astrocytes along with the human GFAP (gfaABC1D) promoter (8), into the mPFC of *Alkbh5^{loxp/loxp}* mice **(Figure 3)**. To address the specificity of AAV-gfaABC1D-eGFP-iCre expression, we immunostained S100 β of the mPFC slice from *Alkbh5^{loxp/loxp}* mice injected with GFAP-icre virus **(Figure 3b)**. Additional quantification was conducted, and the results showed that mostly eGFP-positive cells co-stained with S100 β in the slices **(Extended Data Fig. 8a)**. The mice with the *Alkbh5* knockdown in the mPFC astrocytes also exhibited antidepressant-like effects under stress **(Figure 3)**. Moreover, we injected the DIO-*Alkbh5* virus into the mPFC of *Fgfr3-iCreER^{T2}* transgenic mice, and the results showed that over-expressing astrocytic *Alkbh5* in the mPFC was sufficient to induce depressive-like behaviors **(Figure 3)**.

To rule out the effect of the *Fgfr3-iCre* mouse line in driving recombination in olfactory bulb interneuron, we injected AAV2/9-Syn-eGFP-WPRE-pA virus into the olfactory bulb of *Alkbh5^{loxp/loxp}* mice **(Extended Data Fig. 8)**. Behavioral tests were performed, and results showed that no significant difference was observed in FST, EPM, LD and OFT. These brain region experiments can exclude the influence of olfactory bulb interneuron. The results have been added in the revised **Extended Data Fig. 8** and described on **Page 9 lines 237-246** in the revised manuscript. Thank you again for your rigorous scientific suggestions.

Fig. 2b, c, Representative images (red, ALKBH5; green, S100 β) **(b)** and quantification of the ALKBH5 positive astrocytes **(c)** of cKO and Ctrl mice, Scale bar=500 μ m (left), 20 μ m(Right). n=6 mice per group. Two-sided unpaired t-test. * $p < 0.05$; ** $p < 0.01$; *** $p < 0.001$, n.s., no significance.

Extended Data Fig. 4e. Quantification of the Alkbh5 positive non-astrocytes of Astrocyte cKO and Ctrl mice. n=6 mice per group. Two-sided unpaired t-test. * $p < 0.05$; ** $p < 0.01$; *** $p < 0.001$, n.s., no significance.

Extended Data Fig. 8 Results of behavioral test of Neuron-specific knockout of Alkbh5 in olfactory bulb. **a**, Representative images of AAV-Syn-eGFP-iCre expression in the olfactory bulb of *Alkbh5^{loxP/loxP}* mice (iCre) (green, eGFP). Scale bars=500 μm (left), 25 μm (right). **b**, Western blotting analysis of ALKBH5 in the olfactory bulb of iCre and Ctrl mice, n=4 mice per group. **c-k**, Statistics analysis of iCre and Ctrl mice in FST (**c**), OFT (**d-g**), LD (**h-j**) and EPM (**k**), n=8-12 mice per group. All data are presented the mean \pm SEM. Two-sided unpaired t-test (**b-k**). * $p < 0.05$; ** $p < 0.01$; *** $p < 0.001$, n.s., no significance.

2. Cortical astrocytes are well-known to have extremely low levels of GFAP (Middeldorp et al. Progress in Neurobiology, 2011). However, immunohistochemistry images from Figures 2 and 3 showed significant GFAP staining in mPFC astrocytes even from control mice. This raised a concern of astrocyte reactivity in the mouse models being used that resulted in altered behavioral phenotypes. The authors should provide additional evidence to rule out this possibility.

Answer: We apologize for the misleading images. As you mentioned above, mPFC astrocytes are well-known to have low levels of GFAP, so we selected a local zoomed images, leading you to the misunderstanding of astrocyte activation. Confocal imaging showed that the astrocytic density was low in mPFC, and no significant change in astrocyte morphology and density. Furthermore, we co-stained *Alkbh5* and *S100β* in the mPFC slices of Astrocytic *Alkbh5* cKO mice. Confocal images showed that *S100β* expression was higher than GFAP in the mPFC. And there was no significant change in the morphology of the astrocytes between the astrocytic *Alkbh5* cKO mice and the control littermates. We decided to replace the GFAP staining images with *S100β* staining in **Figure 2**. We hope that the reviewer will agree with our choice.

Fig. 2b, c, Representative images (red, ALKBH5; green, S100β) (**b**) and quantification of the ALKBH5 positive astrocytes (**c**) of cKO and Ctrl mice, Scale bar=500 μm (left), 20 μm(Right). n=6 mice per group. Two-sided unpaired t-test. * $p < 0.05$; ** $p < 0.01$; *** $p < 0.001$, n.s., no significance.

a-b, Representative images (red, ALKBH5; green, GFAP) (**a**) and quantification of the GFAP positive cells (**b**) of cKO and Ctrl mice, Scale bar=25 μm (left), n=3 mice per group. Two-sided unpaired t-test. * $p < 0.05$; ** $p < 0.01$; *** $p < 0.001$, n.s., no significance.

3. It is interesting that *Alkbh5* deletion in astrocytes decreased m6A levels but increased m6A methylation of GLT-1. How did this happen? The authors suggested that observed phenotypes of synaptic transmission (Figure 6), neuronal morphology and neuronal activities (Figure 7) were caused by increased GLT-1 expression in the absence of astrocytic ALKBH5, if correct, then astrocyte-specific knockdown of GLT-1 with RNAi or CRISPR-Cas9 should reverse these effects. The authors should include additional experiments to test their hypothesis.

Answer: We apologize for our mistake. We corrected this error in the revised manuscript as follows: “Astrocytic *Alkbh5* deletion increases m6A levels and alters mPFC epitranscriptome”. To address the m6A levels in Astrocyte *Alkbh5* cKO mice, additional experiments have been performed. Our results showed an increase of total m6A level in Astrocyte cKO mice compared to littermate controls, consistent with m6A modification of GLT-1 (**Figure 4h**).

To determine the role of GLT-1, we specifically knocked down astrocytic GLT-1 using AAV-gfaABC1D-GLT-1-shRNA. GLT-1-shRNA was microinjected into the mPFC of $\text{GFAP}^{\Delta\text{Alkbh5}}$ mice (*Alkbh5*^{loxp/loxp} mice infected with the AAV-gfaABC1D-GFP-iCre virus in the mPFC) (**Extended Data Fig. 10a, b**). The $\text{GFAP}^{\Delta\text{Alkbh5}}$ mice infected with GLT-1-shRNA showed an increase in immobility in the FST (**Figure 5m**) and spent less time in the SI test after CSDS (**Figure 5n**), suggesting that GLT-1-shRNA reversed the antidepressant-like effects of $\text{GFAP}^{\Delta\text{Alkbh5}}$ mice. No differences were observed in EPM, OFT and LD (**Extended Data Fig. 10c-f**). These results suggest that astrocytic *Alkbh5* mediates depression-related behaviors via GLT-1 m6A methylation. The results have been added in the revised **Figure 5, Extended Data Fig. 10** and described on **Page 11-12 lines 308-318** in the revised manuscript.

a, Quantification of m6A levels in mRNA of the mPFC of Astrocyte cKO and Ctrl mice. $n=6$ mice per group. Two-sided unpaired t-test * $p < 0.05$; ** $p < 0.01$; *** $p < 0.001$, n.s., no significance.

Fig. 5n-o, Statistical analysis of $GFAP^{\Delta Alkbh5}$ mice that injected with AAV-shRNA(GLT-1) or AAV-mCherry in FST (**n**), SI test (**o**). One-way ANOVA followed by Bonferroni's test for multiple comparisons (**n**), or Two-way ANOVA (**o**) with Bonferroni's multiple comparisons test. * $p < 0.05$; *** $p < 0.001$; *** $p < 0.001$; n.s., no significance.

Extended Data Fig. 10 Results of behavioral tests in the $GFAP^{\Delta Alkbh5}$ mice injected with GLT-1-shRNA in the mPFC. a, Representative images of

AAV-gfaABC1D-mCherry-shRNA(GLT-1) expression in the mPFC of GFAP^{ΔAlkbh5} mice (green, GFAP; red, GLT-1). Scale bars=500 μm. **b**, Western blotting analysis of GLT-1 in the mPFC of GFAP^{ΔAlkbh5} mice, n=2-3 mice per group. **c**, Total distance in the open field test. n=7 mice per group. **d**, Center time in the open field test. n=6 mice per group. **e**, Time spent in the open arms in the EPM. n=6 mice per group. **f**, Dark duration in the LD. n=6 mice per group. **g**, Time spent in the interaction zone before and after CSDS without target. n=6-8 mice per group. **h-i**, Quantification of mEPSC frequency (**h**) and amplitudes (**i**). n=6-7 cells from 3 individual mice. All data are presented the mean ± SEM. One-way ANOVA (**b-f, h-i**) or Two-way ANOVA (**g**) with Bonferroni's multiple comparisons test. * $p < 0.05$; ** $p < 0.01$; *** $p < 0.001$, n.s., no significance.

4. SAME was applied to enhance m6A modification as well as ameliorate depressive-like phenotypes induced by CSDS. However, direct evidence showing increased m6A methylation by SAME was missing. If overall methylation was indeed increased, does this change also apply to GLT-1?

Answer: We thank the reviewer for raising this point. Our results suggested that total m6A expression was increased after SAME administration to mice (**Figure 8f**). Additional experiments have been performed to address m6A modification of GLT-1. We performed MeRIP-qPCR assay and identified m6A positions on GLT-1 that were significantly increased in SAME group (**Figure 8g**). The results have been added in the revised **Figure 8g** and described on **Page 14-15 lines 401-402** in the revised manuscript.

Fig. 8g, MeRIP-qPCR of GLT-1 m6A levels in the mPFC of C57BL/6J mice treated with SAME (300 μg mL⁻¹) or vehicle. n=3 per group. * $p < 0.05$; ** $p < 0.01$; n.s., no significance.

Minor points:

1. There were misuses of scientific terms. For instance, the trisynaptic circuit refers to a specific neural circuit in the hippocampus, including neurons from three different hippocampal regions. In contrast, tripartite synapses refer to the structures involving astrocytes and neurons.

Answer: We apologize for our mistake. We corrected this error in the revised manuscript on **Page 3 lines 69** as follows: “Astrocytes, major cell types of glial cells, are the key components of blood-brain barrier (BBB) and also wrap around neuronal synapses to form tripartite synapses structures”.

2. Typos and grammatical mistakes were present throughout the manuscript. The authors should check carefully and correct all the mistakes.

Answer: We apologize for our mistake. We check and corrected the typos and grammatical mistakes in the revised manuscript.

3. Providing a full list of genes identified in Figure 4 would be useful to the readers.

Answer: We have uploaded the relevant supplementary tables in the Attachment (**Supplementary Table 2 and Supplementary Table 3**). The relevant sequencing results have also been uploaded to the GSA database (**CRA012006, Shared URL: <https://ngdc.cnecb.ac.cn/gsa/s/dtJ2qcUK>**). We have added it on **Page 47 lines 1205-1209** in the revised manuscript. We hope that the correction will adequately address the reviewer’s concerns.

3. In Figure 5, the authors altered methylation modification sites of GLT-1 with the CRISPR-Cas9 system in cultured astrocytes. However, whether the downregulation of GLT-1 was directly caused by reduced methylation of GLT-1 was unclear.

Answer: Thank you very much for raising this issue. To address this question, we performed a dual luciferase reporter gene assay to further confirm that the GLT-1 m6A sites regulate GLT-1 expression. And the results showed that the luciferase activity was reduced in cells transfected with the SLC1A2 mutant vector compared with the empty vector control. These results suggest that the GLT-1 3’ UTR m6A site can regulate GLT-1 expression. The results have been added in the revised **Figure 5b** and described on **Page 10 lines 270-275** in the revised manuscript.

Fig.5b, Dual-luciferase reporter constructs with the psiCHECK-2 vector contain Renilla luciferase and firefly luciferase driven by two different promoters. The effects of GLT-1 m6A were examined by cotransfection of a dual-luciferase reporter. One-way ANOVA followed by Bonferroni's test for multiple comparisons * $p < 0.05$; *** $p < 0.001$; *** $p < 0.001$; n.s., no significance.

5. Missing X axis label of currents in Figure 6c.

Answer: We apologize for our mistake. We corrected this error in the revised manuscript

Reviewer 2:

In this paper, the authors delve into an exploration of cell types and mechanisms through which RNA modifications contribute to depressive-like behaviors. Their research highlights the prominence of Alkbh5, an RNA demethylase, which exhibits heightened levels in patients diagnosed with major depressive disorder (MDD) as well as in depression-induced mouse models. Notably, Alkbh5 manifests greater susceptibility to stress in astrocytes compared to neurons and endothelial cells. The paper reveals that manipulation of Alkbh5 activity in astrocytes within the medial prefrontal cortex of mice exerts a modulatory influence on depressive-like behaviors. A pivotal finding attributes the primary impact of Alkbh5 to the glutamate transporter GLT-1. The experimental methodologies employed demonstrate a commendable incorporation of cutting-edge techniques. Below, several points and comments are presented for your consideration:

We thank the reviewer for the comments that “The experimental methodologies employed demonstrate a commendable incorporation of cutting-edge techniques”. We are also grateful for the constructive comments. These comments are valuable and helpful, as well as greatly improved

the manuscript. We have read through comments carefully and have made corrections. We hope that the explanation has sufficiently addressed all of the reviewer's concerns.

1. An intriguing observation pertains to the discernible decrease in FTO mRNA in both blood samples from MDD patients and the CSDS mouse model of depression. Despite FTO mRNA elevation in the PFC of Sus and Res mice alike, the pertinence of FTO expression changes in the blood warrants further inquiry. It would be valuable to explore FTO levels in the LPS mouse model of depression, thereby enhancing the comprehensiveness of the investigation.

Answer: Thank you for your comment. In our results, FTO was decreased in both depression patients and mouse peripheral blood (**Figure 1a, c**), and increased in the mPFC of depression mouse models - CSDS and LPS (**Figure 1d, Extended Data Fig. 1g**), with no change in the DLPCF in human brain (**Figure 1b**). It is noteworthy that Alkbh5 mRNA levels were consistently elevated in the peripheral blood of both humans and mice, as well as in brain tissue, which were negatively correlated with the SI ratio in the SI test (**Figure 1e**). So Alkbh5 was selected for further research. Previous studies have also implicated FTO in the pathophysiology of depression (9-12). However, the relationship between FTO and Alkbh5, and how they are involved in the modulation of m6A and depression is a question worthy of investigation. We discuss in the revised manuscript on **Page 16 lines 452-457** as follows: "Previous studies report that Alkbh5 and FTO have close association with MDD and are certain variations of SNP rs12936694 and SNP rs9939609, respectively, in MDD patients. In one study, global m6A methylation is decreased in the whole blood of both human and mice after acute stress and in another study Alkbh5 and FTO as m6A demethylases and Mettl3, Mettl14, and WTAP as m6A methyltransferases are altered in the MDD patients and the depression models."

Extended Data Fig. 1g, qRT-PCR analysis of FTO in the mPFC in the mice treated with LPS.

n=4 mice per group. n=4 mice per group. Two-sided unpaired t-test * $p < 0.05$; ** $p < 0.01$; *** $p < 0.001$, n.s., no significance.

2. A noteworthy concern emerges regarding the *Fgfr3-iCre* line, which possesses the potential to target not only astrocytes but also progenitor cells inclusive of those contributing to adult neurogenesis in the hippocampus (J Neurosc 2021: 2899-2910). Given this, it is prudent to contemplate the possibility that some behavioral effects could potentially stem from the manipulation of *Alkbh5* activity in adult-born neurons.

Answer: Thank you very much for your advice. As previously reported, after tamoxifen administration, ~90% of all protoplasmic and fibrous astrocytes could be labeled in the adult *Fgfr3-iCreER^{T2}* mouse brain (1, 2). This is useful for conditional gene deletion or overexpression. *Fgfr3-iCreER^{T2}* transgenic mice are widely used in astrocyte research (3-6). However, we noticed that *Fgfr3* is expressed by progenitor cells in the postnatal hippocampus (13). Altogether, given the limitation of these cre mice and specific changes in *Alkbh5* in the mPFC, we performed brain-region knockout by injecting the gfaABC1D-iCre virus, which preferentially targets astrocytes along with the human GFAP (gfaABC1D) promoter into the mPFC (8) (**Figure 3**). The mice with the *Alkbh5* knockout in the mPFC also exhibited antidepressant-like effects under stress (**Figure 3**). Moreover, we injected the DIO-*Alkbh5* virus into the mPFC of *Fgfr3-iCreER^{T2}* transgenic mice, and the results showed that overexpressing astrocytic *Alkbh5* in the mPFC was sufficient to induce depressive-like behaviors (**Figure 3**). These brain region experiments can exclude the influence of progenitor cells in the hippocampus.

3. The profound impact of CSDS stress on dendritic spine density within the mPFC stands out; however, the effects on mEPSC frequency appear somewhat subdued, possibly due to the mode of presentation. A suggestion is to illustrate the mEPSC frequency as averages rather than cumulative histograms, potentially providing a clearer representation.

Answer: Thank you very much for your helpful suggestion. We have changed the presentation of the data with mEPSC frequency averages which are more intuitive (**Figure 6**).

4. In Figure 6 and the corresponding text, the assertion is made that "Astrocytic Alkbh5 prevents the disruption of glutamatergic synaptic transmission from social stress." The term "disruption" may not accurately reflect the minor alterations observed in electrophysiological parameters among cKO mice. An alternative description could be more aligned with the observed effects.

Answer: Thank you very much for your advice. We corrected this description in the revised manuscript on **Page 13 lines 347-349** as follows: "These findings implied that knockout of astrocytic Alkbh5 prevent the attenuation of glutamatergic synaptic transmission during chronic stress."

5. An aspect requiring clarification pertains to the duration of SAME-induced antidepressant effects in mice. Is this effect acute in nature or does it span a longer timeframe?

Answer: Thank you very much for your helpful suggestion. We performed the FST after administration of SAME for three weeks. And the results showed that the immobility time of SAME-treated mice was still reduced compared to control mice, suggesting that SAME has long-lasting antidepressant-like effects. The results have been added in the revised **Figure 8i** and described on **Page 15 lines 405** in the revised manuscript.

Fig. 8i, Statistics analysis of C57BL/6J mice treated with SAME or vehicle in FST(3W), n=12 mice per group. Two-sided unpaired t-test. * $p < 0.05$; ** $p < 0.01$; n.s., no significance.

Minor Comments:

1. It appears that the immunostaining for Alkbh5 is weakly present in Figure 2C. Higher magnification pictures and the usage of alternative colors might improve this presentation. Also,

given that GFAP proves suboptimal as an astrocytic marker in the cortex, have the authors explored alternative markers for astrocytes?

Answer: Thank you very much for your helpful suggestion. We co-stained Alkbh5 and S100 β in the mPFC slices of Astrocytic Alkbh5 cKO mice and littermate controls. Confocal imaging and quantification analysis showed that the level of Alkbh5 in S100 β -positive cells was significantly decreased in the astrocytic Alkbh5 cKO mice, suggesting that selective knockout of Alkbh5 in astrocytes. The results have been added in the revised **Figure 2** in the revised manuscript.

Fig. 2b, c, Representative images (red, ALKBH5; green, S100 β) (**b**) and quantification of the ALKBH5 positive astrocytes (**c**) of cKO and Ctrl mice, Scale bar=500 μ m (left), 20 μ m(Right). n=6 mice per group. Two-sided unpaired t-test. * $p < 0.05$; ** $p < 0.01$; *** $p < 0.001$, n.s., no significance.

2. Line 130 contains an inaccuracy. Neuronal and endothelial cell populations are not present in comparable numbers, as indicated.

Answer: We corrected this description in the revised manuscript on **lines 135-136** as follows: “Neurons, astrocytes and endothelial cells are representative cell types in the adult brain”.

Reviewer 3:

In the manuscript, entitled “Cell-specific and sensitive epitranscriptomic m(6)A of stress response in depression,” Guo and colleagues present an intriguing series of results linking alterations in RNA m6A methylation – and its associated enzymatic machinery – in prefrontal cortex to stress-induced physiological and behavioral alterations associated with depression. In profiling the ‘writers’ and ‘erasers’ of m6A in both the periphery and brain of mice exposed to chronic social

stress, as well as in human MDD vs. control samples, they found that ALKBH5, an RNA demethylase of m6A, is increased in its expression (mRNA and protein) in both blood and brain of MDD patients and chronically stressed male mice (in stress-susceptible, but not stress-resilient, animals). Further explorations in vitro comparing cellular stress-induced expression of Alkbh5 in cultured neurons, astrocytes and endothelial cells revealed an astrocytic-specific pattern of upregulation, which was confirmed following social stress in mPFC of mice after sorting for astrocytes specifically. Thus, they next generated mice with conditional deletion of Alkbh5 specifically in astrocytes (vs. neurons or endothelial cells) and found that such deletion only in astrocytes promoted antidepressant-like behaviors (while counterintuitively decreasing m6A epitranscriptome-wide). Bidirectional behavioral regulation was also confirmed via astrocyte specific OE studies and astrocytic restoration of Alkbh5 in KO mice. To next explore the molecular mechanism through which upregulation of Alkbh5 in astrocytes may contribute to stress-induced behavior, the authors performed a series of in vitro and in vivo Ca²⁺ imaging and electrophysiological studies to demonstrate that upregulation of Alkbh5 in response to chronic stress contributes to: increased m6A on GLT-1 mRNA (with increased expression of GLT-1 also observed), increased stress-induced glutamate uptake into astrocytes and reduced sEPSCs and mEPSCs in cortical neurons, thus suggesting that chronic stress functions, at least in part, in mPFC to alter astrocytic glutamate uptake, thereby reducing the physiological activity of glutamatergic neurons, which would be consistent with studies in both mice and humans suggesting aberrant roles for reduced mPFC activity in driving depressive-like behaviors. Overall, this is a rigorous and exciting study that implicates a novel biological process in the regulation of depressive-like states. However, there are a number of concerns that should be addressed prior to publication.

We thank the reviewer for the comments that “Overall, this is a rigorous and exciting study that implicates a novel biological process in the regulation of depressive-like states”. We also appreciate the comments and suggestions concerning our manuscript. The comments and suggestions are very valuable and helpful for improving our manuscript. We have studied the comments carefully and have made corrections which we hope to meet with approval.

1. While the authors confirmed that *Alkbh5* is upregulated specifically in astrocytes vs. neurons or endothelial cells following chronic stress, they should also explore *Alkbh5*'s expression in other glial cells (e.g., microglia, oligodendrocytes), which may also be responsive to stress. This could be done either by performing FACS/FANS or MACS to isolate the other major glial subtypes, followed by qPCR and/or western blotting analyses.

Answer: Thank you for your valuable suggestion. Additional experiments have been performed to address this question. After 10 days of the CSDS paradigm, the adult C57BL/6J mice displayed social avoidance behaviors. Then the microglia and oligodendrocytes in mPFC were isolated with MACS from the mice for qPCR analysis. The result showed that the *Alkbh5* was significantly decreased in the microglia and oligodendrocytes in mPFC of CSDS mice compared to control mice (**Extended Data Fig. 3a-b**). Meanwhile, *Alkbh5* in astrocytes or neurons was significantly increased, with no change in endothelial cells (**Figure 1**). These results suggest that *Alkbh5* responds differently in brain cells under stress. The results have been added in the revised **Extended Data Fig. 3** and described on **Page 6 lines 149-153** in the revised manuscript.

Extended Data Fig. 3 The *Alkbh5* expression in microglia and oligodendrocytes. a-b, qRT-PCR analysis of *Alkbh5* in the mPFC of Sus, Res and Ctrl mice in microglia (a) and oligodendrocytes (b), $n=6$ per group. Cells were separated by MACS. All data are presented the mean \pm SEM. One-way ANOVA with Bonferroni's multiple comparisons test (a and b). * $p < 0.05$; ** $p < 0.01$; *** $p < 0.001$, n.s., no significance.

2. While the links between increased *Alkbh5* expression in astrocytes, GLT-1 m6A (and its increased expression) and disruptions in glutamate signaling in mPFC following chronic stress are very interesting, a couple more experiments need to be performed to definitively link these processes together – a) the authors should provide evidence that astrocytic OE of *Alkbh5*, which

promotes depressive-like behaviors, results in increased GLT-1 m6A and expression, enhanced glutamate uptake into astrocytes and reduced mEPSCs in mPFC neurons (i.e., to further demonstrate the bidirectional relationship; and b) if the experiments in “a” work as expected, the authors should then pharmacologically inhibit GLT-1 under astrocytic *Alkbh5* OE conditions to definitively show that the reductions in neuronal mEPSCs are caused by alterations in the function of GLT-1 specifically.

Answer: We thank the reviewer for detailed review and bringing this to our attention. We apologize for the misleading description. m6A methylation was increased in the mPFC in Astrocytic cKO mice (**Extended Data Fig.4f**). And knockout of *Alkbh5* enhanced glutamate uptake into astrocytes and reduced mEPSCs in mPFC neurons after 10 days CSDS (**Figure 6**).

To determine the role of GLT-1, we specifically knocked down astrocytic GLT-1 using AAV-gfaABC1D-GLT-1-shRNA. GLT-1-shRNA was microinjected into the mPFC of $\text{GFAP}^{\Delta\text{Alkbh5}}$ mice (*Alkbh5*^{loxp/loxp} mice infected with the AAV-gfaABC1D-GFP-iCre virus in the mPFC) (**Extended Data Fig. 10a, b**). The $\text{GFAP}^{\Delta\text{Alkbh5}}$ mice infected with GLT-1-shRNA showed an increase in immobility in the FST (**Figure 5m**) and spent less time in the SI test after CSDS (**Figure 5n**), suggesting that GLT-1-shRNA reversed the antidepressant-like effects of $\text{GFAP}^{\Delta\text{Alkbh5}}$ mice. No differences were observed in EPM, OFT and LD (**Extended Data Fig. 10c-f**). mEPSCs in the pyramidal neurons were also recorded. The results showed that both the frequency and amplitude of mEPSCs were slightly decreased after conditional knockdown of *Alkbh5* in astrocytes, which was reversed by GLT-1-shRNA (**Extended Data Fig.10h-i**). These results suggest that astrocytic *Alkbh5* mediates depression-related behaviors via GLT-1 m6A methylation. The results have been added in the revised **Figure 5, Extended Data Fig. 10** and described on **Page 11-12 lines 308-318** in the revised manuscript.

Fig.5n-o, Statistical analysis of GFAP^{ΔAlkbh5} mice that injected with AAV-shRNA(GLT-1) or AAV-mCherry in FST (n), SI test (o). One-way ANOVA followed by Bonferroni's test for multiple comparisons (n), or Two-way ANOVA (o) with Bonferroni's multiple comparisons test. * $p < 0.05$; *** $p < 0.001$; *** $p < 0.001$; n.s., no significance.

Extended Data Fig. 10 Results of behavioral tests in the GFAP^{ΔAlkbh5} mice injected with GLT-1-shRNA in the mPFC. **a**, Representative images of AAV-gfaABC1D-mCherry-shRNA(GLT-1) expression in the mPFC of GFAP^{ΔAlkbh5} mice (green, GFAP; red, GLT-1). Scale bars=500 μm. **b**, Western blotting analysis of GLT-1 in the mPFC of GFAP^{ΔAlkbh5} mice, n=2-3 mice per group. **c**, Total distance in the open field test. n=7 mice per group. **d**, Center time in the open field test. n=6 mice per group. **e**, Time spent in the open arms in the EPM. n=6 mice per group. **f**, Dark duration in the LD. n=6 mice per group. **g**, Time spent in the interaction zone before and after CSDS without target. n=6-8 mice per group. One-way ANOVA (**b-f**) or Two-way ANOVA (**g**) with Bonferroni's multiple comparisons test. * $p < 0.05$; ** $p < 0.01$; *** $p < 0.001$, n.s., no significance.

Extended Data Fig. 11e-f, Quantification of mEPSC frequency (**e**) and amplitudes (**f**). n=6-7 cells from 3 individual mice. All data are presented the mean \pm SEM. One-way ANOVA (**e-f**) with Bonferroni's multiple comparisons test. * $p < 0.05$; ** $p < 0.01$; *** $p < 0.001$, n.s., no significance.

3. Similarly, the authors should attempt to demonstrate that links between increased *Alkbh5* expression astrocytes and the observed alterations in neuronal mEPSCs are functionally linked to the observed behaviors. This could be done by performing defeat in WT vs. astrocytic KO mice – which leads to reduced glutamate uptake by astrocytes and increased sEPSCs/mEPSCs in the KO animals - coupled to chemogenetic or optogenetic inhibition of glutamatergic neurons in mPFC to see if the positive effects of *Alkbh5* KO in astrocytes can be reversed simply by silencing the neurons that are receiving increased glutamatergic signaling.

Answer: Thank you for these excellent questions. According to the reviewer's suggestion, we performed additional experiments with a chemogenetic approach. We bidirectionally injected the AAV-CamkII α -hM4D(Gi)-mCherry virus into the mPFC of Astrocyte cKO mice to manipulate the activity of mPFC pyramidal neurons. Confocal images showed that hM4Di was expressed in mPFC neurons (**Extended Data Fig. 12a**). Behavioral results showed that chemogenetic inhibition of glutamatergic neurons in the mPFC reversed the antidepressant-like effects in the FST produced by conditional astrocyte knockout of *Alkbh5* (**Extended Data Fig. 12c**). No differences were observed in EPM, and LD (**Extended Data Fig. 12d-g**). These results suggest that astrocytic *Alkbh5* mediates depression-related behaviors via GLT-1 m6A methylation. The results have been added in the revised **Extended Data Fig. 12** and described on **Page 14 lines 377-388** in the revised manuscript.

Extended Data Fig. 12 Results of behavioral tests of inhibition of the activity of glutamatergic neurons. **a**, Representative images of CamkII α -hM4D(Gi)-mCherry expression in the mPFC of Astrocyte cKO mice (red, mCherry). Scale bars=500 μ m. **b**, Representative traces of whole-cell recording in an acute slice from mPFC neurons expressing hM4Di-mCherry after bath application of 1 μ m CNO. **c**, Immobility time in the FST. n=6-8 mice per group; **d**, Dark duration in the LD. n=6-8 mice per group. **e**, Time spent in the open arms in the EPM. n=6-8 mice per group. All data are presented the mean \pm SEM. One-way ANOVA (**c-e**) with Bonferroni's multiple comparisons test. * $p < 0.05$; ** $p < 0.01$; *** $p < 0.001$, n.s., no significance.

4. Given the prevalence of MDD in females vs. males, it would be interesting to know whether stressed females also display such increases in *Alkbh5* expression in mPFC. Robust social defeat protocols now exists for females and could be employed for these assessments.

Answer: Thank you for your valuable suggestion. Following previous studies (14), we treated experimental female mice with the urine of a male CD1 mouse unknown to the resident CD1 aggressors, then exposed the female mice to a different resident aggressive mouse for 5 to 10 minutes each day for 10 days. Following the CSDS paradigm, mPFC tissue was isolated from the mice and used for qPCR analysis. The result showed that the expression of *Alkbh5* in the mPFC of Sus mice was also increased, as in male mice. These results suggest that astrocytic *Alkbh5* mediates depression-related behaviors via GLT-1 m6A methylation. The results have been added in the revised **Extended Data Fig. 1e** and described on **Page 5 lines 122-126** in the revised manuscript.

Extended Data Fig. 1e, Alkbh5 mRNA in the mPFC of female mice after CSDS. n=3 mice per group. One-way ANOVA with Bonferroni's multiple comparisons test. * $p < 0.05$; ** $p < 0.01$; *** $p < 0.001$, n.s., no significance.

5. Are classical antidepressants, such as fluoxetine, sufficient to reverse aberrant increases in Alkbh5 expression in mPFC?

Answer: Thank you for your valuable suggestion. According to the reviewer's suggestion, we performed additional experiments. Adult male C57BL/6J mice were treated with fluoxetine (10 mg/kg/day) for 7 days (15, 16), then mPFC tissue was isolated from the mice for qPCR analysis. The result showed that fluoxetine treatment decreased the Alkbh5 expression in mPFC. However, it is beyond the scope of this paper to determine whether fluoxetine, a classic SSRI reuptake inhibitor, has an effect on m6A modification of RNA and the mechanism involved. This is clearly a very interesting question, and we will be exploring it further in the work that follows.

a, Western blotting analysis of ALKBH5 in the mPFC of Flux and Ctrl mice. n=4 mice per group. Two-sided unpaired t-test. * $p < 0.05$; ** $p < 0.01$; *** $p < 0.001$, n.s., no significance.

6. While the authors have chosen to focus on social interaction and forced swim tests to monitor depressive-like behaviors (as well as additional anxiety related behaviors), given roles for mPFC

in driving anhedonic symptoms in depression, it will be important that additional measure of anhedonia be assessed.

Answer: Thank you for your valuable suggestion. Additional experiments have been performed to address this question. Before CSDS, Astrocyte cKO and Ctrl mice showed no difference in the sucrose preference test. After 10 days CSDS paradigm, the sucrose preference was decreased in Ctrl mice, whereas Astrocyte cKO mice prevented the development of the anhedonic symptoms. The results have been added in the revised **Figure 2f** and described on **Page 7 lines 180-184** in the revised manuscript.

Fig. 2f, Sucrose preference for Astrocyte cKO and Ctrl mice in the SPT. (f, Astrocyte cKO, n=7; Ctrl, n=10). Two-way ANOVA with Bonferroni's multiple comparisons test. * $p < 0.05$; ** $p < 0.01$; *** $p < 0.001$, n.s., no significance.

7. The SAM experiments to induce m6A are highly non-specific to m6A (i.e., they will impact DNA methylation, histone methylation, etc.) or cell-type and do not fit well with the rest of the story, which is focused on cell-type and mechanism specific effects. I would suggest either moving these data to the supplement or removing them from the manuscript altogether.

Answer: Thank you for your valuable suggestion. Alkbh5 acts as a demethylase, reducing RNA m6A modification, whereas SAMe, as a direct methyl donor, leads to increased RNA m6A modification. Positive and negative regulation may better explain role of m6A modification in depression. In addition, as a commonly used dietary supplement, SAMe has the potential to be used as a complementary antidepressant therapy. Therefore, we would like to keep this part of the result in the manuscript. In order to investigate the role of SAMe in the m6A modification of GLT-1, additional experiments were performed. The MeRIP-qPCR assay showed that the m6A positions on GLT-1 were significantly increased in the SAMe treatment group. The results have

been added in the revised **Figure 8g** and described on **Page 14 lines 401-402** in the revised manuscript.

Fig. 8g, MeRIP-qPCR of GLT-1 m6A levels in the mPFC of C57BL/6J mice treated with SAMe ($300 \mu\text{g mL}^{-1}$) or vehicle. $n=3$ per group. * $p < 0.05$; ** $p < 0.01$; n.s., no significance.

8. There are some concerns over the validations of KO vs. OE effects presented, as it appears that these manipulations are not very robust in manipulating *Alkbh5* in the manner in which the authors wish for them to be manipulated (e.g., the KO of *Alkbh5* in astrocytes appears only very weak). Perhaps this is just a technical issue with the manner in which the authors have chosen to validate these KO/OE effects, but as it stands, additional cell-type specific validations see necessary.

Answer: Thank you for your comment. We co-stained *Alkbh5* and *S100 β* in the mPFC slices of Astrocytic *Alkbh5* cKO mice and littermate controls. Confocal imaging and quantification analysis showed that the level of *Alkbh5* in *S100 β* -positive cells was significantly decreased in the astrocytic *Alkbh5* cKO mice, suggesting that selective knockout of *Alkbh5* in astrocytes. The results have been added in the revised **Figure 2** in the revised manuscript.

Fig. 2b, c, Representative images (red, ALKBH5; green, S100 β) (b) and quantification of the ALKBH5 positive astrocytes (c) of cKO and Ctrl mice, Scale bar=500 μm (left), 20 μm (Right).

n=6 mice per group. Two-sided unpaired t-test. * $p < 0.05$; ** $p < 0.01$; *** $p < 0.001$, n.s., no significance.

9. As presented, the differential expression and MeRIP-seq data appear weak, with the results shown only employing nominal p-values of $p < 0.05$. The authors should re-run their analyses using appropriate multiple testing corrections. Additionally, while I understand the author's focus on GLT-1 for the remainder of the manuscript, however, they should attempt to strengthen these analyses and provide a broader overview of the effects being observed. The MeRIP-seq data shown only employing nominal p-values of $p < 0.05$.

Answer: We apologize for our typo mistake. In our study, the differential expression and MeRIP-seq data analysis employed cutoff for $|\log_2(\text{fold_change})| > 2$ and $q\text{-value} < 0.05$. This is a rigorous criterion for sequencing results. We have corrected it on **Page 45 lines 1162-1164 and line 1171** in the revised Method section. And we have uploaded the relevant supplementary tables in the Attachment (**Supplementary Table 2 and Supplementary Table 3**). The relevant sequencing results have also been uploaded to the GSA database (**CRA012006, Shared URL: <https://ngdc.cnbc.ac.cn/gsa/s/dtJ2qcUK>**). We have added it on **Page 47 lines 1205-1209** in the revised manuscript. We hope that the correction will adequately address the reviewer's concerns.

10. Finally, the data presented indicate that Alkbh5 KO decreases m6A in brain, which runs counter to Alkbh5's known role in removing m6A. Do the authors have a reasonable explanation for this?

Answer: We apologize for our mistake. We corrected this error in the revised manuscript as follows: "Astrocytic Alkbh5 deletion increases m6A levels and alters mPFC epitranscriptome". To address the m6A levels in Astrocyte Alkbh5 cKO mice, additional experiments have been performed. Our results showed an increase of total m6A level in Astrocyte cKO mice compared to littermate controls, consistent with m6A modification of GLT-1 (**Figure 4h**).

a, Quantification of m6A levels in mRNA of the mPFC of Astrocyte cKO and Ctrl mice. n=6 mice per group. Two-sided unpaired t-test * $p < 0.05$; ** $p < 0.01$; *** $p < 0.001$, n.s., no significance.

 Xiong Cao, PhD,
 Dept. of Neurobiology,
 Southern Medical University,
 Guangzhou, China.

1. Young, K.M., Mitsumori, T., Pringle, N., Grist, M., Kessaris, N., and Richardson, W.D. 2010. An *Fgfr3-iCreER(T2)* transgenic mouse line for studies of neural stem cells and astrocytes. *Glia* 58:943-953.
2. Hu, N.Y., Chen, Y.T., Wang, Q., Jie, W., Liu, Y.S., You, Q.L., Li, Z.L., Li, X.W., Reibel, S., Pfrieger, F.W., et al. 2020. Expression Patterns of Inducible Cre Recombinase Driven by Differential Astrocyte-Specific Promoters in Transgenic Mouse Lines. *Neurosci Bull* 36:530-544.
3. Leng, L., Zhuang, K., Liu, Z., Huang, C., Gao, Y., Chen, G., Lin, H., Hu, Y., Wu, D., Shi, M., et al. 2018. Menin Deficiency Leads to Depressive-like Behaviors in Mice by Modulating Astrocyte-Mediated Neuroinflammation. *Neuron* 100:551-563 e557.
4. Slezak, M., Kandler, S., Van Veldhoven, P.P., Van den Haute, C., Bonin, V., and Holt, M.G. 2019. Distinct Mechanisms for Visual and Motor-Related Astrocyte Responses in Mouse Visual Cortex. *Curr Biol* 29:3120-3127 e3125.
5. Yu, X., Nagai, J., and Khakh, B.S. 2020. Improved tools to study astrocytes. *Nat Rev Neurosci* 21:121-138.

6. Xiong, W., Cao, X., Zeng, Y., Qin, X., Zhu, M., Ren, J., Wu, Z., Huang, Q., Zhang, Y., Wang, M., et al. 2019. Astrocytic Epoxyeicosatrienoic Acid Signaling in the Medial Prefrontal Cortex Modulates Depressive-like Behaviors. *J Neurosci* 39:4606-4623.
7. Pringle, N.P., Yu, W.P., Howell, M., Colvin, J.S., Ornitz, D.M., and Richardson, W.D. 2003. Fgfr3 expression by astrocytes and their precursors: evidence that astrocytes and oligodendrocytes originate in distinct neuroepithelial domains. *Development* 130:93-102.
8. Cui, Y., Yang, Y., Ni, Z., Dong, Y., Cai, G., Foncelle, A., Ma, S., Sang, K., Tang, S., Li, Y., et al. 2018. Astroglial Kir4.1 in the lateral habenula drives neuronal bursts in depression. *Nature* 554:323-327.
9. Milaneschi, Y., et al.. The effect of FTO rs9939609 on major depression differs across MDD subtypes. *MOL PSYCHIATR* 19, 960-962 (2014).
10. Du, T., et al.. An association study of the m6A genes with major depressive disorder in Chinese Han population. *J AFFECT DISORDERS* 183, 279-286 (2015).
11. Engel, M., et al.. The Role of m6A/m-RNA Methylation in Stress Response Regulation. *NEURON* 99, 389-403 (2018).
12. Liu, S., et al.. Fat mass and obesity-associated protein regulates RNA methylation associated with depression-like behavior in mice. *NAT COMMUN* 12, 6937 (2021).
13. Grońska-Pęski, Marta et al. "Enriched Environment Promotes Adult Hippocampal Neurogenesis through FGFRs." *The Journal of neuroscience : the official journal of the Society for Neuroscience* vol. 41,13 (2021): 2899-2910.
14. Harris, A., Atsak, P., Bretton, Z. et al. A Novel Method for Chronic Social Defeat Stress in Female Mice. *Neuropsychopharmacol.* 43, 1276–1283 (2018).
15. Tseng, YT., Zhao, B., Ding, H. et al. Systematic evaluation of a predator stress model of depression in mice using a hierarchical 3D-motion learning framework. *Transl Psychiatry* 13, 178 (2023).
16. David, D. J. et al. Neurogenesis-dependent and -independent effects of fluoxetine in an animal model of anxiety/depression. *Neuron* 62, 479–493 (2009)

REVIEWER COMMENTS

Reviewer #1 (Remarks to the Author):

Guo et al. performed additional experiments and analyses to address questions raised by the reviewers. The manuscript has improved significantly. However, there are several issues that the authors should carefully address to meet the publication criteria.

Major point:

1. The authors demonstrated that *Alkbh5* deletion in astrocytic *Alkbh5* deletion increased the m6A methylation of GLT-1. Consistently, glutamate release was attenuated in astrocyte *Alkbh5* cKO mice, supported by iGluSnFR imaging (Fig. 5) and the frequency and amplitude of mEPSC were both reduced, supported by electrophysiological recordings (Fig. 6). Given these results, it is irrational to inhibit excitatory neurons with hM4Di DREADD in astrocyte cKO mice (Extended Fig. 12), because (i) the mEPSCs were already diminished compared with control mice, and (ii) action potentials were unaffected in astrocyte cKO mice (Fig. 6). What is even more puzzling is that this further reduction in neuronal activities restored behavioral changes in astrocyte cKO mice. How do the authors explain this?

2. The authors performed immunohistochemistry with S100b in order to address the specificity of astrocyte expression or knockdown of ALKBH5. However, this analysis only confirmed whether astrocytes were targeted but not whether astrocytes were specifically targeted. The authors should provide additional analyses with markers for other cells.

Minor points:

1. In Fig. 1e, expression of ALKBH5 mRNA was similarly increased in control, *Sus*, and *Res* mice. Thus, it does not support the statement "Notably, ALKBH5 mRNA levels..., which were negatively correlated with SI ratios in the SI test (Fig. 1e)." The authors should either revise the text or provide additional analysis to better support their point.

2. In Extended Fig 3, ALKBH5 expression in oligodendrocytes was not labelled as significantly different in *Sus* mice. Did the authors forget to include statistical information on the figure? If there is no statistical difference, then the authors should revise their conclusion that "qRT-PCR analysis showed that ALKBH5 was decreased in microglia and oligodendrocytes in the mPFC of *Sus* mice (Extended Data Fig. 3a-b)."

3. Fig. 2g showed that astrocyte cKO mice spent significantly less time exploring the open arm. But the text said "After mice were exposed to anxiety-related behavioral tests, no behavioral differences were observed in the elevated plus maze (EPM) (Fig. 2g), ..." The authors should correct this.

4. Fig. 3c illustrated western blot results not "Confocal imaging and quantification analysis showed that mostly eGFP positive cells co-stained with S100b in the slices (Fig. 3c)..." Please correct this mistake.

5. The authors found that *Alkbh5* deletion in astrocytes presented anti-depressant effects under chronic social defeat stress and prevented stress-induced excitatory synaptic changes (Fig. 6). However, in the text, they claimed that "ALKBH5 in astrocytes protects glutamatergic synaptic transmission under stress", which is the reverse and illogical.

6. Typos and grammatical mistakes were still present. The authors should check carefully and correct all the mistakes.

Reviewer #2 (Remarks to the Author):

The comprehensive reviews within the manuscript thoroughly address the significant concerns raised by the reviewers. It is evident that the authors have exerted considerable effort to rectify and enhance the data presented in the earlier version of the manuscript.

Reviewer #3 (Remarks to the Author):

The authors have done an outstanding and thorough job at responding to my previous critiques, as well as those raised by the other Reviewers. I continue to find this to be an interesting, well presented and novel study, which will be of great interest to many in the field. I have no further comments and am now supportive of this manuscript's publication at Nature Communications.

Author response

Manuscript ID: NCOMMS-23-32659A

Title: Cell-specific and sensitive epitranscriptomic m(6)A of stress response in depression

We deeply appreciate the valuable suggestions you have provided and the opportunity to improve our manuscript. In response to the reviewer's comments, we have conducted additional experiments to address the majority of the concerns raised. We have revised and corrected the manuscript, taking into account your comments and suggestions. Below, we provide point-by-point responses to the reviewer's comments. We are grateful to all the reviewers for their insightful feedback, which has helped us clarify any ambiguities in the manuscript. We hope that the revised version now better communicates our research findings.

Reviewer #1 (Remarks to the Author):

Guo et al. performed additional experiments and analyses to address questions raised by the reviewers. The manuscript has improved significantly. However, there are several issues that the authors should carefully address to meet the publication criteria.

We are grateful for the constructive comments. These comments are valuable and helpful, as well as greatly improved the manuscript. We have read through comments carefully and have made corrections. We hope that the explanation has sufficiently addressed all of the reviewer's concerns.

Major point:

1. The authors demonstrated that *Alkbh5* deletion in astrocytic *Alkbh5* deletion increased the m6A methylation of GLT-1. Consistently, glutamate release was attenuated in astrocyte *Alkbh5* cKO mice, supported by iGluSnFR imaging (Fig. 5) and the frequency and amplitude of mEPSC were both reduced, supported by electrophysiological recordings (Fig. 6).

Given these results, it is irrational to inhibit excitatory neurons with hM4Di DREADD in astrocyte cKO mice (Extended Fig. 12), because (i) the mEPSCs were already diminished compared with control mice, and (ii) action potentials were unaffected in astrocyte cKO mice (Fig. 6). What is even more puzzling is that this further reduction in neuronal activities restored behavioral changes in astrocyte cKO mice. How do the authors explain this?

Answer: Thank you for your helpful comments. GLT-1, the major glutamate transporter in astrocytes, is responsible for approximately 95% of the total glutamate activity in the mature brain^{1, 2}. In our study, we observed an elevated m6A methylation and expression of GLT-1 in astrocytes from ALKBH5 cKO mice compared to the control group. According to the decay time of mEPSC (Figure 6), Glutamate clearance was increased by astrocytic GLT-1 rather than glutamate release being attenuated in the astrocytic ALKBH5 cKO mice. Previous research has shown that GLT-1 overexpression diminishes the frequency and amplitude of mEPSCs in mPFC pyramidal neurons, without significant alterations in action potentials^{3, 4}. Consistent with these findings, our data

revealed a minor decrease in mEPSC frequency and amplitude in mPFC pyramidal neurons from Astrocyte cKO mice under baseline conditions, potentially attributed to the elevation of GLT-1 expression in astrocytes (Figure 6). The action potential in Figure 6 was assessed under the current-clamp mode without inhibitors were present in the perfusion solution to block glutamatergic and GABAergic synaptic activity. Thus, the action potentials recorded in our study are indeed the result of a combination of factors, encompassing excitatory inputs, and inhibitory inputs, alongside the inherent capability of the cell to generate endogenous action potentials. Notably, we did not observe significant changes in action potentials in ALKBH5 cKO mice, as previous GLT-1 overexpression patch clamp study^{3,4}. And the neuronal morphology and activity in ALKBH5 cKO mice were intact. These results suggest that ALKBH5 knockout in astrocytes may induce excessive glutamate clearance in perisynapse, leading to a decrease in the frequency and amplitude of mEPSCs under physiological conditions, but there was no effect on overall extracellular glutamate levels, neuronal activity or morphology (Figure 5, 6, 7).

It has been reported that excessive glutamate release is triggered by maladaptive stress responses, resulting in excitotoxicity and impairment of synaptic transmission, neuronal morphology and function⁵⁻⁸. These impairments are associated with depressive-like behaviors^{9,10}. In our study, the glutamate levels in the mPFC of susceptible mice were substantially increased after the CSDS. And the frequency of both sEPSCs and mEPSCs in the pramydal neurons of the mPFC of susceptible mice was reduced (Figure 7), as reported in previous studies^{7, 11-13}. Additionally, we also found decreased neuronal spine density and diminished calcium activity following CSDS. GLT-1 is able to remove excess glutamate, and the expression and function of GLT-1 is impaired in depressed patients and in mouse models of depression^{14, 15}. Notably, deletion of GLT-1 in mice has been reported to decrease glutamate uptake, subsequently inducing depressive-like behaviors¹⁵⁻¹⁷. In mice models of depression, direct or indirect increases in GLT-1 expression promote glutamate clearance, protect neuronal function and morphology, and prevent mEPSC changes caused by excessive glutamate release^{3, 4, 13}. In the astrocytic ALKBH5 cKO mice, m6A methylation and expression of GLT-1 in astrocytes were increased (Figure 5). Under stress conditions, upregulated GLT-1 expression in ALKBH5 cKO mice can promote glutamate clearance, counter excessive glutamate-induced changes in the mEPSC, and also protect mPFC pyramidal neuron morphology and Ca²⁺ activity. After CSDS, we found that mPFC neuronal activity was significantly reduced in Ctrl mice during interaction with a CD-1 aggressor in FIT, and stimulus-evoked neuronal activity was significantly increased in astrocyte cKO mice during interaction with a CD-1 aggressor in FIT as before CSDS. We believe that the ALKBH5 cKO mice produce antidepressant-like effects on the basis of the actions described above (Figs. 5, 6, 7). By the way, FST, as an acute intense stress, also leads to increased glutamate release¹⁸⁻²². In the FST, chemical genetic (hM4Di) inhibition of mPFC pyramidal neurons blocked the antidepressant-like effect of ALKBH5 cKO mice (Extended Data Fig. 13). To date, multiple studies have reported that photostimulation of mPFC neurons could produce antidepressant-like effects in FST and TST test²³⁻²⁵, and the photoinhibition of mPFC neurons can induce social avoidance behaviors in social interaction test²⁶. These results suggest that chemical genetic inhibition of mPFC pyramidal neurons in ALKBH5 cKO mice suppresses all of the above protective effects under stress conditions and abolishes the antidepressant effects of ALKBH5 cKO mice.

Although both similar scenarios led to alterations in the frequency and amplitude of neuronal

mEPSC, we believe that the underlying mechanisms differ between them. We have discussed the differences in their molecular mechanisms in the Discussion section of the paper. Meanwhile, the relationship between the two mechanisms remains worthy of further exploration, particularly in terms of its impact on neuron activity and prefrontal cortex circuits in depression.

2. The authors performed immunohistochemistry with S100b in order to address the specificity of astrocyte expression or knockdown of ALKBH5. However, this analysis only confirmed whether astrocytes were targeted but not whether astrocytes were specifically targeted. The authors should provide additional analyses with markers for other cells.

Answer: Thank you very much for your helpful suggestion. We co-stained ALKBH5 and NeuN, Iba1, Sox10 in the mPFC slices of Astrocytic ALKBH5 cKO mice and littermate controls. Confocal imaging and quantification analysis showed that the ALKBH5 positive neurons, oligodendrocytes and microglias were not significantly changed in the astrocytic ALKBH5 cKO mice, suggesting that selective knockout of ALKBH5 in astrocytes. The results have been added in the revised **Extended Data Fig. 5** and described on **Page 7 lines 176-180** in the revised manuscript.

Fig.5 a, Representative images (red, ALKBH5; green, NeuN) (Left) and quantification of the ALKBH5 positive neurons (Right) of Astrocyte cKO and Ctrl mice, Scale bar=20 μ m. n=6 slice from 3 mice per group. **b**, Representative images (red, Sox10; green, ALKBH5) (Left) and quantification of the ALKBH5 positive oligodendrocytes (Right) of Astrocyte cKO and Ctrl mice, Scale bar=25 μ m. n=6 slice from 3 mice per group. **c**, Representative images (red, Iba1; green, ALKBH5) (Left) and quantification of the ALKBH5 positive microglia (Right) of Astrocyte cKO

and Ctrl mice, Scale bar=20 μ m. n=6 slice from 3 mice per group. Two-sided unpaired t-test. n.s., no significance.

Minor points:

1. In Fig. 1e, expression of ALKBH5 mRNA was similarly increased in control, Sus, and Res mice. Thus, it does not support the statement “Notably, ALKBH5 mRNA levels..., which were negatively correlated with SI ratios in the SI test (Fig. 1e).” The authors should either revise the text or provide additional analysis to better support their point.

Answer: Thank you very much for your advice. In Figure 1e, we exhibit an illustrative image depicting the correlation between the SI Ratio and mRNA level, the heatmap in the figure represent the R square (* $p < 0.05$; ** $p < 0.01$). As shown in the depiction, the level of ALKBH5 in the Ctrl, Sus, Res groups is correlated with SI Ratio respectively. However, the expression of ALKBH5 mRNA was not similarly increased in control, Sus and Res mice. This visualization is based on an article published in Nature Neuroscience, providing a comprehensive understanding of the underlying relationships²⁷. Since a negative correlation cannot be visualised using the R square, we corrected this description in the revised manuscript on **Page 5 lines 115-117** as follows: “Notably, ALKBH5 mRNA levels were increased in human and mouse peripheral blood as well as in brain tissues; and the ALKBH5 mRNA levels in the mPFC of Ctrl, Sus and Res mice correlated with SI Ratio in the SI test, respectively (Fig. 1e).”

2. In Extended Fig 3, ALKBH5 expression in oligodendrocytes was not labelled as significantly different in Sus mice. Did the authors forget to include statistical information on the figure? If there is no statistical difference, then the authors should revise their conclusion that “qRT-PCR analysis showed that ALKBH5 was decreased in microglia and oligodendrocytes in the mPFC of Sus mice (Extended Data Fig. 3a-b).”

Answer: Thank you very much for your advice. We corrected this description in the revised manuscript on **Page 6 lines 152-154** as follows: “qRT-PCR analysis showed that ALKBH5 was decreased in microglia in the mPFC of Sus mice and ALKBH5 was no significant change in oligodendrocytes (Extended Data Fig. 3a-b).”

3. Fig. 2g showed that astrocyte cKO mice spent significantly less time exploring the open arm. But the text said “After mice were exposed to anxiety-related behavioral tests, no behavioral differences were observed in the elevated plus maze (EPM) (Fig. 2g), ...” The authors should correct this.

Answer: Thank you very much for your advice. We corrected this description in the revised manuscript on **Page 7 lines 192-195** as follows: “After mice were exposed to anxiety-related behavioral tests, Astrocyte cKO mice spent less time in the open arms of elevated plus maze (EPM) (Fig. 2g), but no behavioral differences were observed in the light-dark box (LD) (Extended Data Fig. 4k)”

4. Fig. 3c illustrated western blot results not “Confocal imaging and quantification analysis showed that mostly eGFP positive cells co-stained with S100b in the slices (Fig. 3c)...” Please correct this mistake.

Answer: Thank you very much for your advice. We corrected this description in the revised manuscript on **Page 8 lines 219** as follows: “Confocal imaging and quantification analysis showed that mostly eGFP positive cells co-stained with S100 β in the slices (Fig. 3b, Extended Data Fig. 7a).”

5. The authors found that Alkbh5 deletion in astrocytes presented anti-depressant effects under chronic social defeat stress and prevented stress-induced excitatory synaptic changes (Fig. 6). However, in the text, they claimed that “ALKBH5 in astrocytes protects glutamatergic synaptic transmission under stress”, which is the reverse and illogical.

Answer: Thank you for your helpful advice. We corrected this description in the revised manuscript on **Page 12 lines 322-323 and Page 31 lines 763-764** as follows: “Knockout of ALKBH5 in astrocytes protects glutamatergic synaptic transmission under stress.”

6. Typos and grammatical mistakes were still present. The authors should check carefully and correct all the mistakes.

Answer: We apologize for our mistake. We check and corrected the typos and grammatical mistakes in the revised manuscript.

Reviewer #2 (Remarks to the Author):

The comprehensive reviews within the manuscript thoroughly address the significant concerns raised by the reviewers. It is evident that the authors have exerted considerable effort to rectify and enhance the data presented in the earlier version of the manuscript.

I am deeply grateful to you and the other reviewers for your positive evaluations and support of this manuscript. I am grateful for your professional guidance and valuable suggestions throughout the review process. Your recommendations have been crucial in elevating the quality and standards of the manuscript.

Reviewer #3 (Remarks to the Author):

The authors have done an outstanding and thorough job at responding to my previous critiques, as well as those raised by the other Reviewers. I continue to find this to be an interesting, well presented and novel study, which will be of great interest to many in the field. I have no further comments and am now supportive of this manuscript's publication at Nature Communications.

I am deeply grateful to you and the other reviewers for your positive evaluations and support of this manuscript. I am grateful for your professional guidance and valuable suggestions throughout the review process. Your recommendations have been crucial in elevating the quality and standards of the manuscript.

References:

1. Abdallah, C.G., *et al.*. Glutamate metabolism in major depressive disorder. *Am J Psychiatry* **171**, 1320-1327 (2014).
2. Danbolt, N.C., Storm-Mathisen, J. & Kanner, B.I. An [Na⁺ + K⁺]coupled l-glutamate transporter purified from rat brain is located in glial cell processes. *NEUROSCIENCE* **51**, 295-310 (1992).
3. Filosa, A., *et al.*. Neuron-glia communication via EphA4/ephrin-A3 modulates LTP through glial glutamate transport. *NAT NEUROSCI* **12**, 1285-1292 (2009).
4. Valtcheva, S. & Venance, L. Astrocytes gate Hebbian synaptic plasticity in the striatum. *NAT COMMUN* **7**, 13845 (2016).
5. Treccani, G., *et al.*. Stress and corticosterone increase the readily releasable pool of glutamate vesicles in synaptic terminals of prefrontal and frontal cortex. *MOL PSYCHIATR* **19**, 433-443 (2014).
6. Öngür, D., Drevets, W.C. & Price, J.L. Glial reduction in the subgenual prefrontal cortex in mood disorders. *Proceedings of the National Academy of Sciences* **95**, 13290-13295 (1998).
7. Ota, K.T., *et al.*. REDD1 is essential for stress-induced synaptic loss and depressive behavior. *NAT MED* **20**, 531-535 (2014).
8. Wang, Y. & Qin, Z. Molecular and cellular mechanisms of excitotoxic neuronal death. *APOPTOSIS* **15**, 1382-1402 (2010).
9. Kang, H.J., *et al.*. Decreased expression of synapse-related genes and loss of synapses in major depressive disorder. *NAT MED* **18**, 1413-1417 (2012).
10. Banasr, M., *et al.*. Glial pathology in an animal model of depression: reversal of stress-induced cellular, metabolic and behavioral deficits by the glutamate-modulating drug riluzole. *Mol Psychiatry* **15**, 501-511 (2010).
11. Li, N., *et al.*. Glutamate N-methyl-D-aspartate receptor antagonists rapidly reverse behavioral and synaptic deficits caused by chronic stress exposure. *Biol Psychiatry* **69**, 754-761 (2011).
12. Hashimoto, K., Sawa, A. & Iyo, M. Increased Levels of Glutamate in Brains from Patients with Mood Disorders. *BIOL PSYCHIAT* **62**, 1310-1316 (2007).
13. Fan, J., *et al.*. O-GlcNAc transferase in astrocytes modulates depression-related stress susceptibility through glutamatergic synaptic transmission. *The Journal of clinical investigation* (2023).
14. Haugseto, Ø., *et al.*. Brain Glutamate Transporter Proteins Form Homomultimers*. *J BIOL CHEM* **271**, 27715-27722 (1996).
15. Fullana, M.N., *et al.*. Regionally selective knockdown of astroglial glutamate transporters in infralimbic cortex induces a depressive phenotype in mice. *GLIA* **67**, 1122-1137 (2019).
16. Rothstein, J.D., *et al.*. Knockout of Glutamate Transporters Reveals a Major Role for Astroglial Transport in Excitotoxicity and Clearance of Glutamate. *NEURON* **16**, 675-686 (1996).
17. Tanaka, K., *et al.*. Epilepsy and Exacerbation of Brain Injury in Mice Lacking the Glutamate Transporter GLT-1. *SCIENCE* **276**, 1699-1702 (1997).
18. Pothula, S., *et al.*. Cell-type specific modulation of NMDA receptors triggers antidepressant actions. *MOL PSYCHIATR* **26**, 5097-5111 (2021).
19. Ma, S., *et al.*. Sustained antidepressant effect of ketamine through NMDAR trapping in the LHb. *NATURE* **622**, 802-809 (2023).
20. Hare, B.D., *et al.*. Optogenetic stimulation of medial prefrontal cortex Drd1 neurons produces rapid and long-lasting antidepressant effects. *NAT COMMUN* **10**, 223 (2019).

21. Duman, R.S., Aghajanian, G.K., Sanacora, G. & Krystal, J.H. Synaptic plasticity and depression: new insights from stress and rapid-acting antidepressants. *NAT MED* **22**, 238-249 (2016).
22. Lowy, M.T., Wittenberg, L. & Yamamoto, B.K. Effect of acute stress on hippocampal glutamate levels and spectrin proteolysis in young and aged rats. *J NEUROCHEM* **65**, 268-274 (1995).
23. Fuchikami, M., *et al.*. Optogenetic stimulation of infralimbic PFC reproduces ketamine's rapid and sustained antidepressant actions. *Proc Natl Acad Sci U S A* **112**, 8106-8111 (2015).
24. Hare, B.D., *et al.*. Optogenetic stimulation of medial prefrontal cortex Drd1 neurons produces rapid and long-lasting antidepressant effects. *NAT COMMUN* **10**, 223 (2019).
25. Liu, J., *et al.*. Astrocyte dysfunction drives abnormal resting-state functional connectivity in depression. *SCI ADV* **8** (2022).
26. Lee, E., *et al.*. Left brain cortical activity modulates stress effects on social behavior. *SCI REP-UK* **5**, 13342 (2015).
27. Dion-Albert, L., *et al.*. Vascular and blood-brain barrier-related changes underlie stress responses and resilience in female mice and depression in human tissue. *NAT COMMUN* **13**, 164 (2022).

REVIEWERS' COMMENTS

Reviewer #1 (Remarks to the Author):

The authors have addressed my concerns and the manuscript has been improved significantly.